# Examining the relationship between intermediate scale soil moisture and terrestrial evaporation within a semi-arid grassland

Raghavendra B. Jana, Ali Ershadi, and Matthew F. McCabe

Water Desalination and Reuse Centre, Division of Biological and Environmental Sciences and Engineering, King Abdullah
University of Science and Technology, Thuwal 23955, Saudi Arabia.

*Correspondence to*: Raghavendra B. Jana (Raghavendra.Jana@kaust.edu.sa)

**Abstract.** Interactions between soil moisture and terrestrial evaporation affect water cycle behaviour and responses between the land surface and the atmosphere across scales. With strong heterogeneities at the land surface, the inherent spatial variability in soil moisture makes its representation via point-scale measurements challenging, resulting in scale-mismatch
when compared to coarser-resolution satellite-based soil moisture or evaporation estimates. The Cosmic Ray Neutron Probe (CRNP) was developed to address such issues in the measurement and representation of soil moisture at intermediate scales. Here we present a study to assess the utility of CRNP soil moisture observations in validating model evaporation estimates. The CRNP soil moisture product from a pasture in the semi-arid central-west region of New South Wales, Australia was compared to evaporation derived from three distinct approaches, including the Priestley-Taylor (PT-JPL), Penman-Monteith
(PM-Mu) and Surface Energy Balance System (SEBS) models, driven by forcing data from local meteorological station data and remote sensing retrievals from the Moderate Resolution Imaging Spectroradiometer (MODIS) sensor. Pearson's Correlations, Quantile-Quantile (Q-Q) plots, and Analysis of Variance (ANOVA) were used to qualitatively and quantitatively evaluate the temporal distributions of soil moisture and evaporation over the study site. The relationships were examined against nearly two years of observation data, as well as for different seasons and for defined periods of analysis. Results
highlight that while direct correlations of raw data were not particularly instructive, the Q-Q plots and ANOVA illustrate that the root-zone soil moisture represented by the CRNP measurements and the modelled evaporation estimates reflect similar distributions under most meteorological conditions. The PT-JPL and PM-Mu model estimates performed contrary to expectation when high soil moisture and cold temperatures were present, while SEBS model estimates displayed a disconnect from the soil moisture distribution in summers with long dry spells. Importantly, no single evaporation model matched the
statistical distribution of the measured soil moisture for the entire period, highlighting the challenges in effectively capturing evaporative flux response within changing landscapes. One of the outcomes of this work is that the analysis points to the feasibility of using intermediate scale soil moisture measurements to evaluate gridded estimates of evaporation, exploiting the independent, yet physically linked nature of these hydrological variables.

**Keywords**: CRNP*; soil moisture; land surface evaporation; SEBS; PM-Mu; PT-JPL*

## 1. Introduction

Land surface evaporation and soil moisture play major roles in defining the water cycle behaviour of landscapes as well as controlling the feedback from the land surface to the atmosphere at a range of spatial and temporal scales (Manfreda et al. 2007; Seneviratne et al. 2010). The coupling between soil moisture and the overlaying atmosphere has been a topic of intense investigation in recent years. Koster et al. (2006) compared multiple atmospheric general circulation models with regard to the strength of land-atmosphere couplings and reported that while the coupling strengths varied widely for the models, most models agreed upon certain locations of high land-atmosphere coupling. Dirmeyer (1994) used a simple biosphere model to evaluate the effect of soil moisture and vegetation stress on the climatology of drought, while Martens et al. (2016) showed that assimilating satellite- derived soil moisture improved model estimates of terrestrial evaporation at the continental scale. Land-atmosphere coupling studies have also investigated, among others aspects, the impact of soil moisture on precipitation (Eltahir 1998; Koster et al. 2004; Schär et al. 1999) and how this knowledge can be an indicator of climate change (Seneviratne et al. 2006); the links between soil moisture and cloud cover (Betts 2004); and also how the ENSO cycle influences the coupling and the surface-atmosphere feedbacks (Miralles et al. 2014).

Land surface evaporation (sometimes also referred to as ET) comprises the processes of plant transpiration, evaporation from the soil and evaporation from canopy intercepted rainfall (Kalma et al. 2008), and has been estimated to return up to 70% of precipitated water back to the atmosphere (Hanson 1991; Trenberth et al. 2011). In arid and semi-arid regions, this value can be much larger. Although coupling between evaporation and soil moisture is expected to be high in arid and semi-arid regions, the dynamics of surface-atmosphere feedbacks are not well understood in such environments (Wang et al. 2012). Several studies have attempted to describe these links, with the aim of predicting one variable through knowledge of the other (Mintz and Walker 1993; Wetzel and Chang 1987) or to use developed relationships to inform upon linked hydrological responses such as evaporation (McCabe et al. 2005; Stisen et al. 2011), soil moisture (Liu et al. 2012), drought (Entekhabi et al. 1992; Fischer et al. 2007; Oglesby and Erickson 1989), precipitation (Findell et al. 2011; Held et al. 2005), and even vegetation response to soil moisture stress (Liu et al. 2011). A common feature of such studies is the use of model estimates of terrestrial evaporation across a wide range of study areas and land cover and biome types. The reliance on these model estimates necessitates a more critical examination of both the models used and an evaluation of their performance.

With this in mind, the Global Energy and Water Cycle Exchanges (GEWEX) LandFlux project (McCabe et al. 2016) and the related Water Cycle Multi-mission Observation Strategy − Evapotranspiration (WACMOS-ET) project (Michel et al. 2016; Miralles et al. 2016) reflect ongoing efforts to develop strategies for the prediction of land surface fluxes at regional and global scales. As part of these activities, studies were undertaken to compare the remote sensing derived evaporation products with tower-based measurements: a standard approach to flux evaluation (Ershadi et al. 2014). However, such comparisons suffer from both spatial and temporal scale mismatches, making robust evaluations inherently challenging. Generally, the spatial

footprint of satellite-based sensors is much larger than the fetch of an eddy covariance tower. Furthermore, while tower-based sensors routinely record information at intervals of between 15-30 minutes throughout the diurnal cycle, many satellite observations used in hydrological studies are generally instantaneous retrievals that may only be available at a daily interval. While questions on the suitability of comparing large scale gridded evaporation estimates to fine scale tower observations have

been raised previously (McCabe et al. 2016), they remain largely unresolved. As such, identifying complimentary observation sources that can be used to improve upon the evaluation of a variety of hydrological processes is a much needed objective (McCabe et al. 2008). This critical need forms a key motivation of this work where we look for an answer to the question: are independent hydrological data-sets available that can be used to inform upon linked elements of the hydrological cycle?

From an observational perspective, a range of approaches have been employed to obtain soil moisture values at multiple resolutions (Jana and Mohanty 2012; Jana et al. 2008; Vereecken et al. 2007). Generally, soil moisture measurements are made using either in-situ devices that are ground-based or via air- and satellite-borne sensors. While in-situ measurements tend to represent a spatial scale on the order of centimetres, remotely sensed soil moisture products have resolutions on the order of several hundred meters (airborne) to tens of kilometres (satellite based) (Vereecken et al. 2007). Unfortunately, field scale

spatial variability of soil moisture is generally smoothed out in large scale soil moisture estimates (Manfreda et al. 2007). While technically a number of point-scale measurements can be collected and then spatially averaged over a domain, such an approach is infeasible for continuous monitoring over multiple fields. Establishment of critical zone observatories in recent years has provided valuable insight into hydrological process due to their intensive instrumentation (Lin et al. 2011). However, they involve significant outlays involved with regards to finance, physical effort, and time. Although such observatories are

invaluable in providing data to understand the underlying processes, it remains impractical to implement a large number of sensors across any and every field of interest. As such, there is a clear need for resolving field scale heterogeneity at scales that can be monitored from remotely observing platforms or conversely, providing ground based observations that better reflect the scale of satellite systems.

In recent years, the challenge of obtaining intermediate resolution (between point scale and satellite resolution) soil moisture has been addressed by the use of Cosmic Ray Neutron Probes (CRNP) (Zreda et al. (2012). Based on determining the neutron density of cosmic rays, CRNP are able to make measurements of soil moisture at spatial resolutions of a few hundred meters. The CRNP footprint, with a radius of approximately 130 m (tropical climate) to 240 m (arid/semi-arid climates) (Kohli et al. 2015), is comparable to that of airborne remote sensors, while being ground-based and capable of continuously recording soil

moisture over long periods of time. Moreover, the effective measurement depth of the CRNP sensor ranges from 12 to 76 cm, depending on the degree of saturation. This depth allows the sensor to capture the root zone soil moisture dynamics to a great extent (Desilets et al. 2010).

Development of such approaches to capture the field scale dynamics of soil moisture brings with it the ability to more closely explore the interactions between the surface and atmosphere, and their linking mechanisms. Soil surface evaporation and plant transpiration processes are influenced significantly by the root zone soil moisture: transpiration rates depend upon the amount of plant-available water in the root zone, while evaporation plays a regulatory role in governing the dynamics of the surface soil moisture. It is well recognized that soil moisture is a limiting factor in the evaporative process (Seneviratne et al. 2010), playing an important role in modulating plant stress and vegetation response. With improved sensing of the root-zone soil moisture, it is expected that any modelled relationship between evaporation and soil moisture will be more robust. From an observational standpoint, however, it has been challenging to explore these links directly due to the mismatch in data scales. Using CRNP soil moisture data collocated with gridded model estimates of evaporation may provide some insight into these processes and relationships.

Given this, the objective of this study is to investigate the potential of using CRNP soil moisture retrievals to evaluate modelled evaporation estimates derived from a combination of tower based and remote sensing inputs. The capacity to indirectly monitor surface flux responses using such data offers a mechanism through which land-atmosphere couplings can be explored and provides an additional constraint on coupled water and energy cycle modelling at the land surface. In order to achieve this objective, we conducted an exploratory study that examined the link between the soil moisture retrievals from CRNP and evaporation estimates collected over a semi-arid grassland. The underlying hypothesis to be tested here is that rainfall input in such landscapes will be well reflected by the soil moisture values, which in turn should be strongly coupled to evaporation.

## 2. Data and Methodology

### 2.1. Study area

The study was conducted over a pasture site near Baldry, a rural township in the central-west of New South Wales, Australia. The site is classified as a semi-arid region with latitude -32.87 degrees, longitude 148.52 degrees, and an elevation of 438 m above mean sea level. The Baldry experimental catchment formed part of the Australian National Cosmic Ray Soil Moisture Monitoring Facility (CosmOz) network, established by the Commonwealth Scientific and Industrial Research Organization (CSIRO) (Hawdon et al. 2014). A CNRP was put in place at the site in March 2011, complimenting existing meteorological and soil moisture sensors, as well as an eddy-covariance flux tower. Figure 1 details the location of the Baldry test site. Further details regarding the instrumentation at the Baldry test site and the CosmOz network in general are provided by Hawdon et al. (2014) (see also http://cosmoz.csiro.au/sensor-information/?SiteNo=1).

### 2.2. Description of evaporation models

#### 2.2.1 PT-JPL

Of the three evaporation models evaluated in this study, the Priestley-Taylor Jet Propulsion Laboratory (PT-JPL) model uses the least number of meteorological and remote sensing input data, including air temperature, humidity, net radiation and a vegetation index (Fisher et al. 2008). The model has been used to estimate actual evapotranspiration at local (Ershadi et al. 2014) and global scales in various studies (e.g., Badgley et al. 2015; Ershadi et al. 2014; Vinukollu et al. 2011), including the recent LandFlux (McCabe et al. 2016) and WACMOS-ET (Miralles et al. 2016) efforts. Detailed descriptions of the model are provided in those references. The PT-JPL is a three-source model that uses net radiation ($R_n$), normalized difference vegetation index (NDVI), air temperature and humidity as inputs. Total evaporation is partitioned into soil evaporation ($\lambda E_s$), canopy transpiration ($\lambda E_t$), and wet canopy evaporation ($\lambda E_i$), i.e. $\lambda E = \lambda E_s + \lambda E_t + \lambda E_i$. The model initially partitions the net radiation into soil and vegetation components and then estimates the potential evaporation from the three (soil, canopy and wet canopy) sources. The effects of green cover fraction, relative wetness of the canopy, air temperature, plant water stress and soil water stress on the evaporative process are represented by corresponding multipliers. Actual evaporation values for each component of the system are computed by reducing the potential evaporation based on these constraint multipliers. Further details of the PT-JPL model are given in Appendix A at the end of this article and also in the work of Fisher et al. (2008).

#### 2.2.2 PM-Mu

The Penman-Monteith based model developed by Mu et al. (2011) (PM-Mu) is another three-source scheme that has been used in a range of applications for estimating terrestrial fluxes (Mu et al. 2013), including forming the basis behind the global evaporation product (MOD16) (Mu et al. 2013). The PM-Mu model computes total evaporation as the sum of the three components: soil evaporation, canopy transpiration and evaporation of the intercepted water in the canopy, i.e. ($\lambda E = \lambda E_s + \lambda E_t + \lambda E_i$). Inputs to the model include net radiation ($R_n$), normalized difference vegetation index (NDVI), air temperature and humidity, and vegetation phenology. While based on the Penman-Monteith equation (Monteith 1965), the evaporation from each component is estimated by assigning weights to the fractional vegetation cover, relative surface wetness and available energy. Plant phenology and climatological data are used to extend biome-specific conductance parameters from the stomata to the canopy scale. The extended conductance parameters are then used to parameterize aerodynamic and surface resistances for each component source. Further details regarding the PM-Mu model can be found in Appendix B below, and also in the publications by Mu et al. (2011) and Mu et al. (2013), along with recent evaluation studies from the local (Ershadi et al. 2014; Michel et al. 2016) to the global scales (McCabe et al. 2016; Miralles et al. 2016).

### 2.2.3 SEBS

The Surface Energy Balance System (SEBS) model (Su 2002a) is a physically-based scheme that has been widely used in estimating evaporation across a range of scales (Elhag et al. 2011; McCabe and Wood 2006; Su et al. 2005a). The model utilizes commonly available hydrometeorological variables, including net radiation, land surface temperature, air temperature, humidity, wind speed and vegetation phenology and height to calculate both latent and sensible heat surface fluxes. The model first calculates land surface roughness parameters, including roughness lengths for momentum and heat transfer using a method developed by Su et al. (2001). These roughness parameters are then applied to a set of flux-gradient equations along with temperature gradient and wind speed data to compute the sensible heat flux. The flux-gradient equations quantify the heat transfer between the land surface and the atmosphere. SEBS uses either the Monin-Obukhov Similarity Theory (MOST) or the Bulk Atmospheric Similarity Theory (BAST) equations (Brutsaert 2005), based on the height of the atmospheric boundary layer. SEBS then determines the sensible heat flux under hypothetical extreme wet and dry conditions to calculate the evaporative fraction. Latent heat flux is estimated as a component of the available energy based on the calculated evaporative fraction. Further details regarding the SEBS model and its formulation can be found in the works of Su (2002a) and Ershadi et al. (2013b). A summarized version is presented in Appendix C at the end of this article.

### 2.3. Datasets

Following is a brief review of the data that has been used in the analysis, as well as a description of the standardization process employed to allow comparison of these distinct datasets.

### 2.3.1 Eddy-covariance surface fluxes

An eddy covariance system provided measurement of heat fluxes and radiation components for periods throughout the duration of the CRNP installation. The system comprised of a Campbell Scientific 3D sonic anemometer (CSAT-3, Campbell Scientific, Logan, UT, USA) along with a LiCOR 7500 (Li-7500, LiCor Biosciences, Lincoln, NB, USA) for high-frequency water vapour and $CO_2$ concentrations. Turbulent flux data was sampled at 10 Hz, with flux values averaged to 30 minute intervals. The height of the eddy covariance tower was fixed at approximately 2 m, providing an estimated fetch of approximately 200 m (Leclerc and Thurtell (1990). A meteorological tower was co-located alongside the eddy covariance system, with a Kipp and Zonen CNR4 radiometer, Apogee infrared surface temperature, RIMCO rain gauge, Vaisala HMP75C temperature and humidity probe, RM Young wind sentry (wind speed and direction), Huskeflux ground heat flux plate and Vaisala BaroCap barometric pressure sensor. Both tower meteorological and eddy-covariance data were quality controlled to detect and remove errors. The low-frequency 30 minute resolution data were corrected for coordinate rotation (Finnigan et al. 2003) and WPL effects (Leuning 2007) using the PyQC software tool (available from code.google.com/p/eddy). These data were then accumulated to form 24-hourly totals for each day of the available observation period.

### 2.3.2 Satellite-based observations

Information required for the different evaporation models (see Section 2.2) were obtained using both the tower-based observations as well as the Moderate-Resolution Imaging Spectroradiometer (MODIS) sensor on board NASA's Terra and Aqua satellites. Land surface temperature data required for the SEBS model were derived from the daily MOD11A1 and MYD11A1 products of the Terra and Aqua satellites (Wan 2009). Normalized Difference Vegetation Index (NDVI) data (used by all models) were obtained from the MOD13Q1 product (Solano et al. 2010).

### 2.3.3 Soil Moisture

The intermediate scale soil moisture data used in this study was obtained from the COSMOS repository (http://cosmos.hwr.arizona.edu/Probes/StationDat/078/). The level 4 dataset for the soil moisture was chosen from the repository, since corrections for atmospheric pressure and water vapour variation as well as for incoming flux neutron density due to location and the presence of other hydrogen sources were already incorporated in the data as per the methods presented by Hawdon et al. (2014). The CRNP device was active at the Baldry site for the period March 30, 2011 to March 13, 2014. Accounting for gaps in the dataset due to instrument outage as well as the availability of concurrent ancillary data such as the remote sensing inputs required for the evaporation models provided a total of 684 days of data from 2011-2013 that were used for the study. The 15-minute temporal frequency of CRNP soil moisture estimates were averaged to the daily time scale. In addition to the CRNP data, soil moisture observations from three water content reflectometers (CS616, Campbell Scientific, Logan, UT, USA) installed at the site were also used. The TDR sensors had probe lengths of 30 cm that were inserted vertically from the surface at three locations around the CRNP instrument. An average of the three TDR measurements was computed at each time step for comparison with the CRNP soil moisture estimates.

### 2.4. Data Standardization

Volumetric soil moisture obtained via the CRNP is reported in units of $cm^3.cm^{-3}$ and represents a fraction that is always less than a theoretical maximum of 1. Evaporation estimates are represented in units of mm and have a less defined theoretical upper bound. In order to compare these quantities across different units and ranges, evaporation and soil moisture data were standardized by computing the standard score, i.e.,

$$X_{st} = \frac{X - \bar{X}}{\sigma} \tag{1}$$

where $X_{st}$ is the standardized data point, $X$ is the raw data, $\bar{X}$ is the mean of the raw dataset, and $\sigma$ is the standard deviation of the raw dataset. Reducing data with different units to units of standard deviation helps to compare the distributions of dissimilar quantities. Standardizing raw data before comparison of distributions with different means and scales of variation is a common practice (Koster et al. 2011; Zhang et al. 2008). Importantly, doing so does not change the statistical analyses, as the data are simply scaled to be comparable to each other.

## 3. Analyses and Discussion

### 3.1. Correlation of CRNP soil moisture and evaporation at the EC tower

Given that modelled evaporation estimates are generally validated against observations from eddy covariance towers (Ershadi et al. 2014), an initial step in our study was to query the relationship between the CRNP soil moisture retrieval and the tower-based evaporation observations. Soil moisture and land surface evaporation are both processes with inherent stochasticity in their determination, due in part to the imprecise nature of physical measurement i.e. field-based observations of these processes are inferred rather than measured in an absolute sense. In such situations, it is often more appropriate to analyse the quantities for their similarity of statistical distributions rather than deducing a lack (or presence) of a relationship based on point-to-point statistics such as correlation. Figure 2a shows a scatter of the raw (non-standardized) CRNP soil moisture daily averages plotted against the corresponding 24-hour daily evaporation values observed at the eddy covariance tower on site. The two quantities have a Pearson's correlation coefficient close to 0.40. Figure 2b shows the scatter of change in soil moisture on a daily scale versus the corresponding change in observed evaporation across a daily interval for the entire period of record at the Baldry site. The plot is clustered around the zero change values, with a correlation coefficient of -0.26. These low correlations suggest that the CRNP soil moisture and the observed evaporation are statistically not well correlated with each other. However, comparing the cumulative distribution (CDF) of the change in soil moisture state at each daily time step to the CDF of the corresponding change in observed evaporation (Figure 2d) demonstrates a visible similarity between the two quantities. Physically, it makes sense that the change in evaporation should be correlated to change in soil moisture, even if the raw data does not show that response clearly. The strength of the relationship is brought out at the distribution level, whereas the point-to-point comparison fails to do so. The CDF's of the raw data (Figure 2c) also show that the quantities behave similar to each other, albeit with a shift in scale.

### 3.2. Analyses of standardized data distributions

Figure 3 shows the precipitation data at the study site, together with the standardized CRNP soil moisture and evaporation estimates from the PT-JPL, PM-Mu and SEBS models. The average of the three TDR soil moisture measurements, and the evaporation observed from the in-situ eddy covariance tower, are also shown. As can be seen, the CRNP and TDR soil moisture series reflect similar responses in their variability and seasonality, with the CRNP data indicating greater fine-scale variability. While it might be expected that the TDR data should display greater variability in response, the CRNP measurements have higher variability in soil moisture values. This could be due to factors such as the variability in the measurement depth of the CRNP with change in the saturation, and higher sensitivity of the CRNP to near-surface moisture as compared to deeper layers (Bogena et al. 2013). High frequency variations at the soil surface may also be attenuated in the TDR signal since it is integrated over the 30 cm probe depth.

With the aim of investigating the link between the CRNP soil moisture and the modeled evaporative response, non-parametric quantile-quantile (Q-Q) plots and box plots were used. Figure 4 shows the Q-Q plots for the standardized soil moisture data versus the standardized model derived evaporation estimates. The inter-quartile range is highlighted in grey and the red line denotes the extrapolation of the slope of the inter-quartile range. Good agreement between the Q-Q plot and the red expectation line indicates that the two quantities have been sampled from the same distribution. It can be seen that for all three models, the evaporation and soil moisture relationships follow the expectation (red line) closely, particularly in the inter-quartile range. This indicates that the two independent datasets (modelled evaporation and CRNP soil moisture) are sampled from distributions that are very similar to each other. However, beyond the inter-quartile range, a deviation of the plot away from the expectation line is seen in all cases. This is more apparent at the extremes. The SEBS estimates deviate more from the expectation on the higher side of the inter-quartile range, as compared to the other two models. However, this behaviour is similar to that of the measured evaporation from the eddy covariance tower, as can be seen from the last plot in Figure 4. Towards the higher end of the data range, the PT-JPL and PM-Mu values, while close to the expectation, are underestimated when compared to the tower measurements.

The above observation is additionally supported by the boxplots resulting from the one-way Analysis of Variance (ANOVA) test, which are shown in Figure 5. In general, an ANOVA test generates a box plot along with a table of statistics, the most important of which is typically the p-value. The p-value, ranging between 0 and 1, signifies how dissimilar the average values of two datasets are from each other. A high p-value signifies that the two averages are statistically similar to each other, and vice versa. However, for this study, the datasets were standardized to have zero-mean distributions in order to enable comparison between quantities having different units and ranges. Hence, in this case, the ANOVA test would always return a p-value of 1, indicating that the two means are statistically identical. The p-value is, therefore, not suitable to understand the behaviour of the two datasets. Hence, the ANOVA boxplots are preferred in this case as they provide insight into the distribution-level behaviour of the datasets.

From Figure 5 it can be seen that the CRNP soil moisture distribution is slightly skewed to the left (shorter tail below the first quartile), while the evaporation model estimates are more symmetrically distributed. Again, as in the Q-Q plots, the inter-quartile ranges are very similar for both hydrological variables. In this case, the SEBS evaporation estimates have more outliers on the higher side, as compared to the other two model estimates.

### 3.3. Analysis of relationship based on defined periods

In order to get a better understanding of the observed dynamics between the soil moisture and evaporation signatures, we divided the data time series into four distinct sub-periods based on short-term trends and the level of correspondence between the soil moisture and evaporation records. Period 1 ran from day of record (DoR) 55 to 144. In the course of this period both the soil moisture and the modelled evaporation had a steady descending tendency (see Figure 3). The soil moisture and

evaporation signatures behaved consistent with each other in both increasing and decreasing tendencies during Period 2, running from DoR 321 to 410. Period 3, between DoR 410 and DoR 500, covers a series of wetting and drying cycles of soil moisture, while no such corresponding changes were observed in the evaporation signatures. Period 4, covering the period between DoR 501 and DoR 590, exhibits signatures that oppose each other in behaviour such that an increase in soil moisture

corresponds to a decrease in the estimated evaporation. The four sub-periods were selected to examine the relationship between the CRNP soil moisture and the model derived evaporation estimates under diverse hydrological conditions. Data from these sub-periods were individually standardized using the mean and standard deviation for each particular period.

Q-Q plots of CRNP soil moisture against the model-derived evaporation estimates for each of the four sub-periods are shown

in Figure 6. The corresponding Pearson's correlation value (R) for each of the modelled evaporation estimates with regard to the CRNP soil moisture is also included. In these analyses, the R-value is used to assess the relative performance of the models during each specific period, and not as an absolute indicator. During sub-periods 1 and 2, the Q-Q plots indicate that the soil moisture and evaporation model-estimates were both sampled from similar distributions, particularly in the inter-quartile range (IQR). Quantile values of the standardized PT-JPL model estimates have the closest match with the soil moisture values during

Period 1, both within and outside the IQR. This observation is supported by the PT-JPL estimate having a higher correlation (0.108) with the soil moisture among the three model estimates. During Period 2, the PM-Mu model estimates display a higher correlation (0.764) with the soil moisture than any of the other two models. However, if we limit the analysis to the IQR data, where the bulk of the data resides, the SEBS model (R=0.714) has a closer match to the expectation line. For Period 3, a simple visual analysis of the standardized time series of soil moisture and evaporation would indicate that the two quantities were

probably de-coupled. However, the Q-Q plots for this period display that there is substantial correspondence between the two datasets. This demonstrates that although the changes in the evaporation were small when compared with the soil moisture variability for this period, evaporation was still likely being driven by the root zone soil moisture. Also, while the scales are shifted, the Q-Q plots suggest that the evaporation values are still being sampled from a distribution similar to that of the soil moisture. The PT-JPL model estimates track the expectation line closely within the IQR, and on the higher extremes beyond

the IQR. Below the IQR, all three models behave consistently with each other, suggesting that as the soil gets drier, the three models studied here all converge towards similar estimates of evaporation. During this period, negative correlations with the soil moisture are observed for all three model evaporation estimates.

It is possible that under certain conditions of vegetation type and density, the CRNP measurement might include canopy

intercepted water along with the soil moisture. Canopy interception in densely vegetated surfaces could also be a significant contributor to the total evaporative flux. In such cases, the TDR measurements of soil moisture may better represent the root zone wetness conditions. However, in the semi-arid grassland environment of the current study, canopy interception is unlikely to comprise a significant component of the terrestrial evaporation. In order to test if the TDR measurements were more representative of the root zone soil moisture, we performed a Q-Q analysis using the average of the three TDR measurements

at the study site instead of the CRNP measurements. We found that there was no significant difference in the plots, and thus these results are not reported here.

The Q-Q plots for sub-period 4 show that there is a marked incongruity between the distributions of soil moisture and the PT-JPL evaporation estimates. The plot deviates from the line representing the slope of the IQR, particularly within IQR where the bulk of the data would be expected to lie. This behaviour indicates that during this sub-period, the modelled evaporation was de-coupled from the soil moisture signature. The other three-source model (PM-Mu), also exhibits this de-coupling, although to a lesser extent. However, the de-coupling is not seen in the flux estimates of the SEBS model. The PT-JPL and PM-Mu estimates also show negative correlation with the soil moisture during this period, while the SEBS estimate is positively correlated. Period 4, corresponding with the Australian winter (May-July 2012), received frequent rainfall events (Figure 3) that resulted in elevated soil moisture levels while the cloud cover probably limited the energy available for the evaporation process.

Comparisons between the ANOVA boxplots of the model-derived evaporation to the CRNP soil moisture and the observed flux from the eddy covariance tower, for each period of the trends-based analysis, are shown in Figure 7. During Period 4, the modelled evaporation distributions are significantly different from the eddy-covariance measured at the tower, especially for the PT-JPL and PM-Mu models. The observed evaporation has a distribution that is closer to that of the CRNP soil moisture, while the distributions of the PT-JPL and PM-Mu estimates are skewed to the left. The PT-JPL and PM-Mu methods are based primarily on the available energy (Rn-G) of the system, with soil moisture being implicitly accounted for by adjusting the air humidity. The Penman (1948) and Penman-Monteith (Monteith 1965) combination equations that form the theoretical basis for the PT-JPL and PM-Mu models, were developed for and tested (Rana and Katerji 1998; Shahrokhnia and Sepaskhah 2011; Sumner and Jacobs 2005) in situations where energy limitations were not present. In contrast, the SEBS model follows a more physically-based approach dependent on the turbulent mixing theory (Brutsaert 2013), which is valid in energy-limited situations similar to those observed during Period 4. It is also likely that low temperatures and additional hydro-meteorological factors could have caused a de-coupling of the soil moisture from the air humidity. Due to the low temperatures, the air humidity would be lower, while the frequent precipitation ensures high soil moisture content. This creates a steep gradient for the moisture at the soil-air interface. In such a scenario, regardless of the presence of abundant soil moisture for evaporation, the models which use air humidity (or vapour pressure) as a surrogate for soil moisture may report lower estimates compared to those observed from the eddy-covariance tower. These factors may represent physical constraints on the application of the PT-JPL and PM-Mu methods and require further investigation.

### 3.4. Analysis of relationship based on seasonal behaviour

To understand the seasonal patterns in the relationship between the CRNP soil moisture and the evaporation data, the two-year record of data was partitioned according to seasons. The period between December and February corresponds to the Austral

summer, while autumn is from March to May, winter from June to August, and spring from September to November. The summer and spring seasons experienced the greatest number of precipitation events (defined here as rainfall greater than 1 mm/day) with 34 and 33 rainy days out of a total of 164 and 181 days of records respectively, followed by winter (26 events in 184 days) and autumn (17 events in 155 days).

Corresponding Q-Q plots for the four seasons are shown in Figure 8. In the autumn, the PM-Mu evaporation estimates correspond most closely with the CRNP soil moisture retrievals in the inter-quartile range (IQR), followed by the SEBS estimates. PT-JPL performed the poorest. Beyond the IQR, and overall, the SEBS estimates were the closest match to the soil moisture distribution in this season. In winter, the PT-JPL estimates were the closest to the soil moisture distribution within the IQR, while overall the SEBS model again performed best, relative to these specific metrics. In the spring, the SEBS model estimates were distributed most similar to the soil moisture, both within the IQR and overall. PM-Mu estimates were the least similarly distributed.

The summer season shows that all three evaporation estimates depart from the expectation within the IQR, with the SEBS estimates being least similar and the PT-JPL estimates most similar to the soil moisture distribution. As mentioned above, the site experienced 34 rainfall events out of a total of 164 days of record. However, there were also long periods with no rainfall events. Combined with the higher temperatures of summer, this leads to greater non-monotonic variations in the soil moisture signature, thus creating a disconnect with the evaporation patterns. There are more switches between moisture-constrained and energy-constrained conditions during this season. It has been demonstrated previously that the occurrence of hot and dry periods leads to de-coupling of soil moisture and evaporation (Pollacco and Mohanty 2012). The soil moisture profile in such situations becomes heterogeneous in that the process driving the surface soil moisture variability (mainly soil evaporation) no longer influences the deeper layer soil moisture variability (mainly due to transpiration). Further explanation of this de-coupling process can be found in Pollacco and Mohanty (2012). Evaporation variability in summer is driven more by the precipitation patterns than the soil moisture. With an abundance of energy, and severe limitation of soil moisture, any influx of moisture due to precipitation is quickly evaporated back to the atmosphere. Despite this, in an example of the "correlation does not imply causation" maxim, it is observed that the evaporation estimates for this season exhibit the highest correlation (R-value) with the CRNP soil moisture. High soil moisture and temperature conditions in summer could also increase uncertainty in the surface-to-air temperature gradient: a key element of the SEBS approach. The SEBS model has been shown to be highly sensitive to the land surface temperature gradient parameterization (Ershadi et al. 2013a), and this might be a reason for its poor performance in the summer period.

From Figure 8, it is also seen that in most cases, the PT-JPL and PM-Mu models underestimate the evaporation at the higher end of the scale, at least when compared with the eddy-covariance tower measurements. The SEBS model generally performs better in this regard, as also seen in the previous analysis of the shorter time series (Figure 5). Previous studies have shown

that SEBS based evaporation estimates were found to correspond well with tower-based measurements when there is a short, homogeneous canopy (McCabe et al. 2016) and the Baldry grassland site meets this criterion.

In this study we exploit the physical mechanism that makes soil moisture a key driver of the evaporation process. As such, it makes perfect sense to evaluate models using observations that govern or influence that process to a significant extent. The evaporation models examined here do not make use of soil moisture as an input, and neither do the majority of the satellite based evaporation models, in general. Hence, the evaporation and soil moisture datasets are statistically independent, although physically linked. Given the general lack of observation data concerning any specific process, it is important that independently observed, yet physically linked variables, be used to aid in the evaluation process, as demonstrated by the results of this study.

### 3.5. Caveats, limitations and suggestions for future direction

Correlation analyses are based on one-to-one comparison between datasets. Q-Q plots, on the other hand are a measure of the similarity of distribution. While there may be low point-to-point correlation between two datasets, it is very much possible that the two quantities are sampled from the same (or similar) distributions (Jana et al. 2008). Such correspondence at the distribution level, rather than at the point level, is much more meaningful for stochastic variables such as soil moisture and evaporation. Hence, we emphasize the agreement in the Q-Q plots rather than the R value in our study. This behaviour also emphasizes the need to develop suitable statistical metrics which do not rely upon the traditionally used point to point matches to examine and quantify relationships between stochastic datasets.

Other potential causes of errors could be uncertainties in the observed data from the CRNP instrument and the eddy-covariance, cloud cover resulting in inaccurate MODIS observations which could further lead to inconsistencies in the model outputs. Additionally, uncertainties in the meteorological forcings and other model inputs have not been explored in this study. The model structure and variations in model parameterization could also affect the analyses. We have used the structure and parameterization described by Ershadi et al. (2014) as they have been shown to correspond well with the tower observations. Other model parameterizations may improve (or degrade) the correspondence with the soil moisture, but that investigation is beyond the scope of this preliminary study.

This study shows that intermediate resolution soil moisture can be used to validate and constrain models for land surface evaporation. Importantly, the soil moisture distribution can act as a guide for validating the model evaporation estimates in cases where eddy covariance data is either unavailable or of poor quality. Further, considering that the footprint of the tower observations is at a much finer scale in comparison with the gridded model estimates of evaporation, it may be more prudent to evaluate evaporation models using the CRNP soil moisture, which is at a comparable resolution. Obviously, further analyses across different biomes and hydroclimatic regimes is necessary before a robust relationship between the model evaporation

estimates and the CRNP soil moisture can be established. However, the outcome of this study encourages such an effort to be made. As shown in earlier studies to validate gridded evaporation products (Ershadi et al. 2014; McCabe et al. 2016) no single evaporation model consistently performed better than others across all conditions, whether that was seasonal or based upon climate or biome-type. This suggests that an ensemble modelling approach with model weights assigned according to, among other factors, their established relationship with soil moisture may be more suitable.

## 4.   Conclusions

Relationships between soil moisture observations from a CRNP sensor and evaporation estimates derived from three distinct model structures using a combination of tower and satellite-based data were examined across a semi-arid grassland site. Standardized daily evaporation and CRNP soil moisture data were compared, with an analysis performed over different hydrological regimes, as well as an examination of seasonal scale variations. As theorised, the two hydrological variables displayed significant correspondence with each other over the entire time series, indicating that there is a strong and hydrological consistent connection relating them. It was also established that a relationship exists between the intermediate scale soil moisture measurements and the modelled evaporation estimates across most of the defined analysis periods. It was observed that the PT-JPL and PM-Mu model estimates behaved contrary to expectation in conditions where high soil moisture existed with colder temperatures. SEBS model estimates presented a similar disconnect from the soil moisture distribution, but in the summer season during long dry spells. These deviations are attributed to the model structures and reflect previous works identifying the geographic and temporal variability of model performance. Overall, no single model estimate of evaporation fully reproduced the CRNP soil moisture across all conditions. However, the outcome of this study indicates that the intermediate scale soil moisture could be employed as a useful constraint to validate gridded evaporation estimates derived from models.

## 5.   Acknowledgements

Research reported in this publication was supported by the King Abdullah University of Science and Technology (KAUST), Saudi Arabia. The CosmOz instrument was supported by the Commonwealth Scientific and Industry Research Organization (CSIRO). Instrumentation at the Baldry site was funded and commissioned as part of the Australian government's National Collaborative Research Infrastructure Strategy (NCRIS) and the University of New South Wales. Dr Ershadi was supported by the Australian Research Council Discovery Project (DP120104718).

## Appendix A: Priestley-Taylor Jet Propulsion Laboratory (PT-JPL) model description

In the PT-JPL model (Fisher et al. 2008), total evapotranspiration is partitioned into canopy transpiration ($\lambda E_c$), soil evaporation ($\lambda E_s$) and wet canopy evaporation ($\lambda E_{wc}$) defined as follows:

$$\lambda E_c = k_c \times \alpha_{PT} \frac{\Delta}{\Delta + \gamma} R_n^c$$

$$\lambda E_s = k_s \times \alpha_{PT} \frac{\Delta}{\Delta + \gamma} (R_n^s - G_0) \qquad \text{A1}$$

$$\lambda E_{wc} = k_{wc} \times \alpha_{PT} \frac{\Delta}{\Delta + \gamma} R_n^c$$

where $\alpha_{PT}$ is the Priestley-Taylor coefficient (=1.26); $\Delta$ is the slope of the saturation vapour pressure (kPaK$^{-1}$); $\gamma$ is the psychrometric constant (kPaK$^{-1}$); $R_n^c$ is the net radiation for canopy, $R_n^c = R_n - R_n^s$; $R_n^s$ is the net radiation for soil given by $R_n^s = R_n \exp(-0.6 LAI)$; and $G_0$ is the ground heat flux (Wm$^{-2}$). Total evapotranspiration is then $\lambda E = \lambda E_c + \lambda E_s + \lambda E_{wc}$.

$k_c$, $k_s$ and $k_{wc}$ are reduction functions for scaling of potential evapotranspiration in each of canopy, soil and wet canopy components to their actual values and are defined as:

$$k_c = (1 - f_{wet}) f_g f_T f_M$$

$$k_s = f_{wet} + f_{SM}(1 - f_{wet}) \qquad \text{A2}$$

$$k_{wc} = f_{wet}$$

where $f_{wet}$ is relative surface wetness; $f_g$ is green canopy fraction; and $f_T$ is air temperature constraint. $f_M$ and $f_{SM}$ are empirical factors used as a proxy for plant and soil water stress, respectively. The functions are defined as:

$$f_{wet} = RH^4$$

$$f_g = \frac{f_{APAR}}{f_{IPAR}}$$

$$f_T = \exp\left[-\left(\frac{T_a - T_{opt}}{T_{opt}}\right)^2\right] \qquad \text{A3}$$

$$f_M = \frac{f_{APAR}}{f_{APAR_{max}}}$$

$$f_{SM} = RH^{VPD}$$

where $RH$ represents the relative humidity (fraction); $f_{APAR}$ and $f_{IPAR}$ are fractions of the photosynthesis active radiation ($PAR$) that is absorbed ($APAR$) and intercepted ($IPAR$) by green vegetation cover, defined as $f_{APAR} = 1.3632 \times SAVI - 0.048$ and $f_{IPAR} = NDVI - 0.05$. The optimum plant growth temperature ($T_{opt}$) is the air temperature ($T_a$) that occurs when the canopy activity is the highest, i.e. when the $f_{APAR}$, radiation is at the peak and $VPD$ is at the minimum value. $VPD$ is vapour pressure deficit in kPa. $SAVI$ is the soil adjusted vegetation index, calculated as $SAVI = 0.45 \times NDVI + 0.132$. The leaf area index, $LAI$, used in computation of $R_n^s$, is calculated as $LAI = -\ln(1 - f_c)/k_{PAR}$ with $k_{PAR} = 0.5$ and $f_c = f_{IPAR}$.

## Appendix B: PM-Mu Model description

In the PM-Mu model, total evaporation can be accounted as the sum of the evaporation from the intercepted water in the wet canopy ($\lambda E_{wc}$), from water transpired from the leaves ($\lambda E_t$), and from soil evaporation ($\lambda E_s$). Detailed formulation and parameterization of each of the components as presented by Mu et al. (2011) and Ershadi et al. (2015) are summarized as follows:

**Evaporation of intercepted water from a wet canopy** ($\lambda E_{wc}$) is calculated using the following equation:

$$\lambda E_{wc} = f_w \frac{\Delta A_c + f_c \rho c_p (e^* - e)/r_a^{wc}}{\Delta + \gamma \frac{r_s^{wc}}{r_a^{wc}}} \tag{B1}$$

where $A_c$ is the available energy for the canopy transpiration defined as $A_c = f_c R_n$ and $f_c$ is fractional vegetation cover. $f_w$ is the relative surface wetness, calculated as $f_w = RH^4$, with the formulation developed by Fisher et al. (2008). The aerodynamic resistance $r_a^{wc}$ and surface resistance $r_s^{wc}$ for wet canopy are defined as:

$$r_a^{wc} = \frac{r_h^{wc} r_r^{wc}}{r_h^{wc} + r_r^{wc}} \tag{B2}$$

$$r_s^{wc} = \frac{1}{f_w g_e LAI} \tag{B3}$$

where $r_h^{wc}$ is wet canopy resistance to sensible heat transfer and $r_r^{wc}$ is the wet canopy resistance to radiative heat transfer, defined as following:

$$r_h^{wc} = \frac{1}{f_w g_h LAI} \tag{B4}$$

$$r_r^{wc} = \frac{\rho c_p}{4 \sigma T_a^3}$$

$g_e$ and $g_h$ are leaf conductance to evaporated water vapor and sensible heat (respectively) per unit $LAI$, with biome specific values presented in Table B1. $T_a$ is air temperature (°C) and $\sigma$ is the Stefan-Boltzmann constant.

**Transpiration from the canopy** $(\lambda E_t)$ in the PM-Mu model is calculated as:

$$\lambda E_t = (1 - f_w) \frac{\Delta A_c + f_c \rho c_p (e^* - e)/r_a^t}{\Delta + \gamma \left(1 + \frac{r_s^t}{r_a^t}\right)} \qquad \textbf{B5}$$

where $r_a^t$ is the aerodynamic resistance and $r_s^t$ is the surface resistance for transpiration. The bulk canopy resistance $(r_s^t)$ in the model is formulated as the inverse of the bulk canopy conductance $(C_c)$ and calculated as $r_s^t = \frac{1}{C_c}$. The model assumes that the stomatal conductance $(G_s^{st})$ and cuticular conductance $(G_s^{cu})$ are in parallel, but both are in series with the canopy boundary-layer conductance $G_s^b$. As such, the canopy conductance to transpiration is calculated as:

$$C_c = \begin{cases} (1 - f_w) \dfrac{(G_s^{st} + G_s^{cu}) G_s^b}{G_s^{st} + G_s^{cu} + G_s^b} LAI & , LAI > 0, (1 - f_w) > 0 \\ 0 & , LAI = 0, (1 - f_w) = 0 \end{cases} \qquad \textbf{B6}$$

where $G_s^b = g_h$, $G_s^{cu} = r_{corr} g_{cu}$ and $G_s^{st} = c_L m(T_{min}) m(VPD) r_{corr}$ with $VPD$ being the vapor pressure deficit (Pa).

$g_{cu}$ is the leaf cuticular conductance per unit LAI (assumed equal to 0.00001 m.s$^{-1}$ for all biomes). Also, $c_L$ is the mean potential stomatal conductance per unit leaf area, and is assumed constant for each biome . The $r_{corr}$ is the correction factor for $G_s^{st}$ to adjust it based on the standard air temperature and pressure (20 °C and 101,300 Pa) using the following equation:

$$r_{corr} = \frac{1}{\frac{101300}{Pa} \left(\frac{T_a + 273.15}{293.15}\right)^{1.75}} \qquad \textbf{B7}$$

$m(T_{min})$ is a multiplier that limits potential stomatal conductance by minimum air temperature $(T_{min})$, and $m(VPD)$ is a multiplier used to reduce the potential stomatal conductance when $VPD = e^* - e$ is high enough to reduce canopy conductance. Following Mu et al. (2007), $m(T_{min})$ and $m(VPD)$ are calculated as following:

$$m(T_{min}) = \begin{cases} 1 & T_{min} \geq T_{min}^{open} \\ \dfrac{T_{min} - T_{min}^{close}}{T_{min}^{open} - T_{min}^{close}} & T_{min}^{close} < T_{min} < T_{min}^{open} \\ 0 & T_{min} \leq T_{min}^{close} \end{cases} \qquad \textbf{B8}$$

$$m(VPD) = \begin{cases} 1 & VPD \leq VPD_{open} \\ \dfrac{VPD_{close} - VPD}{VPD_{close} - VPD_{open}} & VPD_{open} < VPD < VPD_{open} \\ 0 & VPD \geq VPD_{close} \end{cases} \qquad \text{B9}$$

Values of $T_{min}^{open}$, $T_{min}^{close}$, $VPD_{open}$ and $VPD_{close}$ are listed in the works of Mu et al. (2011) and Ershadi et al. (2015) for each biome type. The aerodynamic resistance to canopy transpiration, $r_a^t$, is calculated using two parameters: the convective heat transfer resistance $r_h$ and radiative heat transfer resistance $r_r$, by assuming they are in parallel (Thornton 1998). The model uses the following equation to calculate $r_a^t$:

$$r_a^t = \frac{r_h^t r_r^t}{r_h^t + r_r^t} \qquad \text{B10}$$

In this formula, $r_h^t = 1/g_{bl}$ and $r_r^t = r_r^{wc}$ and $g_{bl}$ is the leaf-scale boundary layer conductance per unit *LAI*. Here, the $g_{bl}$ is assumed to be equal to that of the sensible heat (i.e. $g_{bl} = g_h$).

**Evaporation from the soil** ($\lambda E_s$) in the PM-Mu model is based on the sum of evaporation from wet soil ($\lambda E_{ws}$) and evaporation from saturated soil ($\lambda E_{ss}$), calculated using the following equation:

$$\lambda E_s = \lambda E_{ws} + \lambda E_{ss}. \qquad \text{B11}$$

To determine the fractions of wet and saturated soil components, the PM-Mu model uses the relative surface wetness parameter
$f_w$. As such, the evaporation from the wet soil is:

$$\lambda E_{ws} = f_w \frac{\Delta A_s + (1 - f_c)\rho c_p (e^* - e)/r_a^s}{\Delta + \gamma \dfrac{r_s^s}{r_a^s}}. \qquad \text{B12}$$

where $A_s$ is the available energy for soil evaporation calculated as $A_s = (1 - f_c)R_n - G_0$. Evaporation from the saturated soil is calculated as:

$$\lambda E_{ss} = RH^{VPD/\beta}(1 - f_w)\frac{\Delta A_s + (1 - f_c)\rho c_p (e^* - e)/r_a^s}{\Delta + \gamma \dfrac{r_s^s}{r_a^s}} \qquad \text{B13}$$

where $r_a^s$ and $r_s^s$ are resistance parameters for aerodynamic transfer of evaporation from the soil surface to the atmosphere. The $RH^{VPD/\beta}$ term in the above equation is a constraint parameter for soil moisture, with $\beta$ assigned a constant value of 200.
The soil surface resistance $r_s^s$ is calculated as:

$$r_s^s = r_{corr} r_{totc} \qquad \text{B14}$$

where $r_{totc}$ is a function of *VDP* and biological parameters $r_{bl}^{min}$ and $r_{bl}^{max}$ as follows:

$$r_{totc} = \begin{cases} r_{bl}^{max} & VPD \leq VPD_{open} \\ r_{bl}^{max} - \dfrac{(r_{bl}^{max} - r_{bl}^{min}) \times (VPD_{close} - VPD)}{VPD_{close} - VPD_{open}} & VPD_{open} < VPD < VPD_{close} \\ r_{bl}^{min} & VPD \geq VPD_{close} \end{cases} \qquad \textbf{B15}$$

$VPD_{open}$ is the $VPD$ for when leaves transpire with no water stress and $VPD_{close}$ is the $VPD$ when there is no transpiration due to water stress. Values for $r_{bl}^{max}$, $r_{bl}^{min}$, $VPD_{open}$ and $VPD_{close}$ for various land covers are provided in the works of Mu et al. (2011) and Ershadi et al. (2015).

The aerodynamic resistance at the soil surface ($r_a^s$) is parallel to both the resistance to convective heat transfer ($r_h^s$) and the resistance to radiative heat transfer $r_r^s$, with its components calculated as:

$$r_a^s = \frac{r_h^s r_r^s}{r_h^s + r_r^s} \qquad \textbf{B16}$$

where $r_r^s = r_r^{wc}$ and $r_h^s = r_s^s$.

## Appendix C: SEBS Model Description

The SEBS model includes routines for calculating the sensible heat flux ($H$) using meteorological and land surface data, and using $H$ to estimate the latent heat flux as a fraction of the total available energy at the surface. $H$ estimation in the SEBS model follows the physically-based flux-gradient functions of momentum and heat transfer near the surface. When the measurement height of meteorological variables is in the atmospheric surface layer, the SEBS model uses the flux-gradient functions of the Monin-Obukhov similarity theory (MOST) (Monin and Obukhov 1945), as following:

$$u_a = \frac{u_*}{\kappa}\left[\ln\left(\frac{z - d_0}{z_{0m}}\right) - \Psi_m\left(\frac{z - d_0}{L}\right) + \Psi_m\left(\frac{z_{0m}}{L}\right)\right] \qquad \textbf{C1}$$

$$\theta_s - \theta_a = \frac{H}{\kappa u_* \rho c_p}\left[\ln\left(\frac{z - d_0}{z_{0h}}\right) - \Psi_h\left(\frac{z - d_0}{L}\right) + \Psi_h\left(\frac{z_{0h}}{L}\right)\right] \qquad \textbf{C2}$$

where $z$ is the measurement height for the meteorological variables (m), $u_a$ is wind speed (m.s$^{-1}$), $u_*$ is the friction velocity (m.s$^{-1}$) , $\rho$ is the density of the air (kg.m$^{-3}$), $c_p$ is specific heat capacity of air at constant pressure (J.kg$^{-1}$.K$^{-1}$), $\kappa$ (= 0.41) is the von Karman's constant (-), $\theta_s$ is the potential land surface temperature (K), $\theta_a$ is the potential air temperature (K) at height $z$, $H$ is the sensible heat flux (W.m$^{-2}$) , $d_0$ is the zero-plane displacement height (m), $z_{0m}$ is the roughness height for momentum transfer (m), $z_{0h}$ is the roughness height for heat transfer (m) and $\Psi_m$ and $\Psi_h$ are the stability correction functions for momentum and heat transfer. $L$ is the Obukhov length (m) defined as:

$$L = -\frac{\rho c_p u_*^3 \theta_v}{\kappa g H}$$

**C3**

with $g$ the acceleration due to gravity (m.s$^{-2}$) and $\theta_v$ the atmospheric virtual potential temperature (K).

SEBS uses the stability-correction functions proposed by Beljaars and Holtslag (1991) for stable conditions and the functions proposed by Brutsaert (2005) are used for unstable conditions. The roughness length for momentum and heat transfer ($z_{0m}$ and $z_{0h}$) are estimated using the methodology developed by Su et al. (2001), which employs vegetation phenology, air temperature and wind speed.

SEBS uses a correcting method to scale the MOST derived sensible heat flux between hypothetical dry and wet limits based on the relative evapotranspiration concept. Finally, this scaled sensible heat flux can be used to calculate the evaporative fraction ($\Lambda$), which then can be used to calculate the latent heat flux as $\lambda E = \Lambda(\mathrm{R_n} - G_0)$. Further details on the SEBS model description are provided by Su (2002b) and Su et al. (2005b).

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

**Figures**

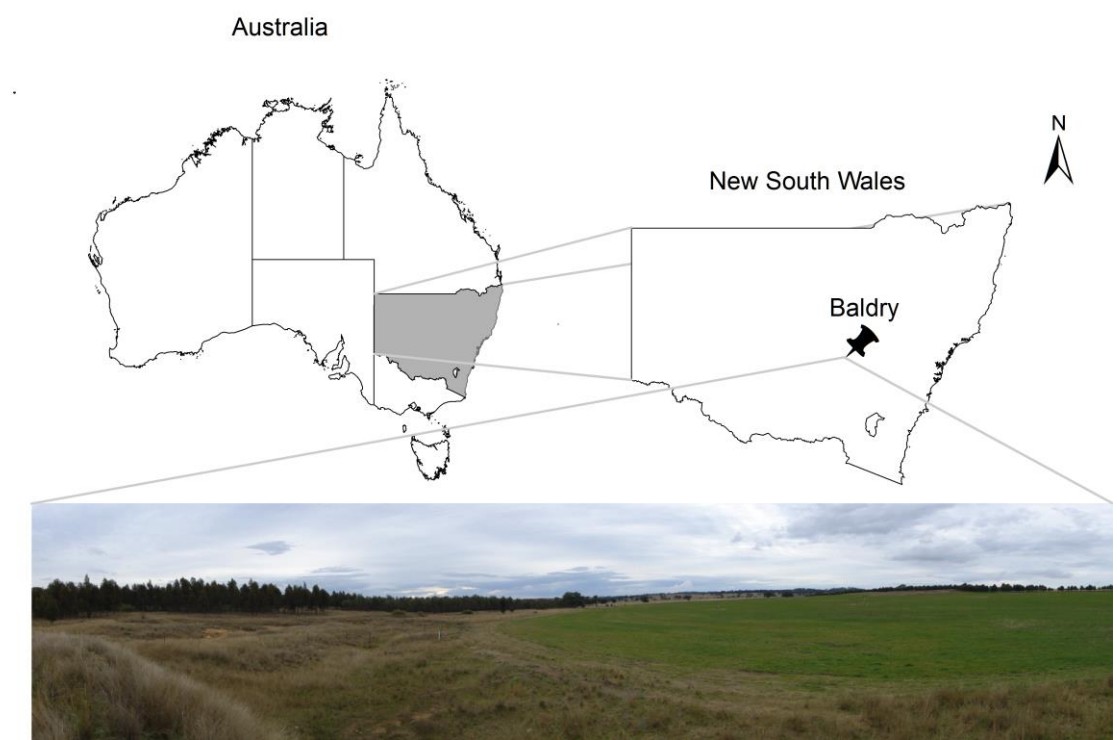

**Figure 1: Location of the Baldry study area in the central-west of NSW Australia, along with a photograph of the study area.**

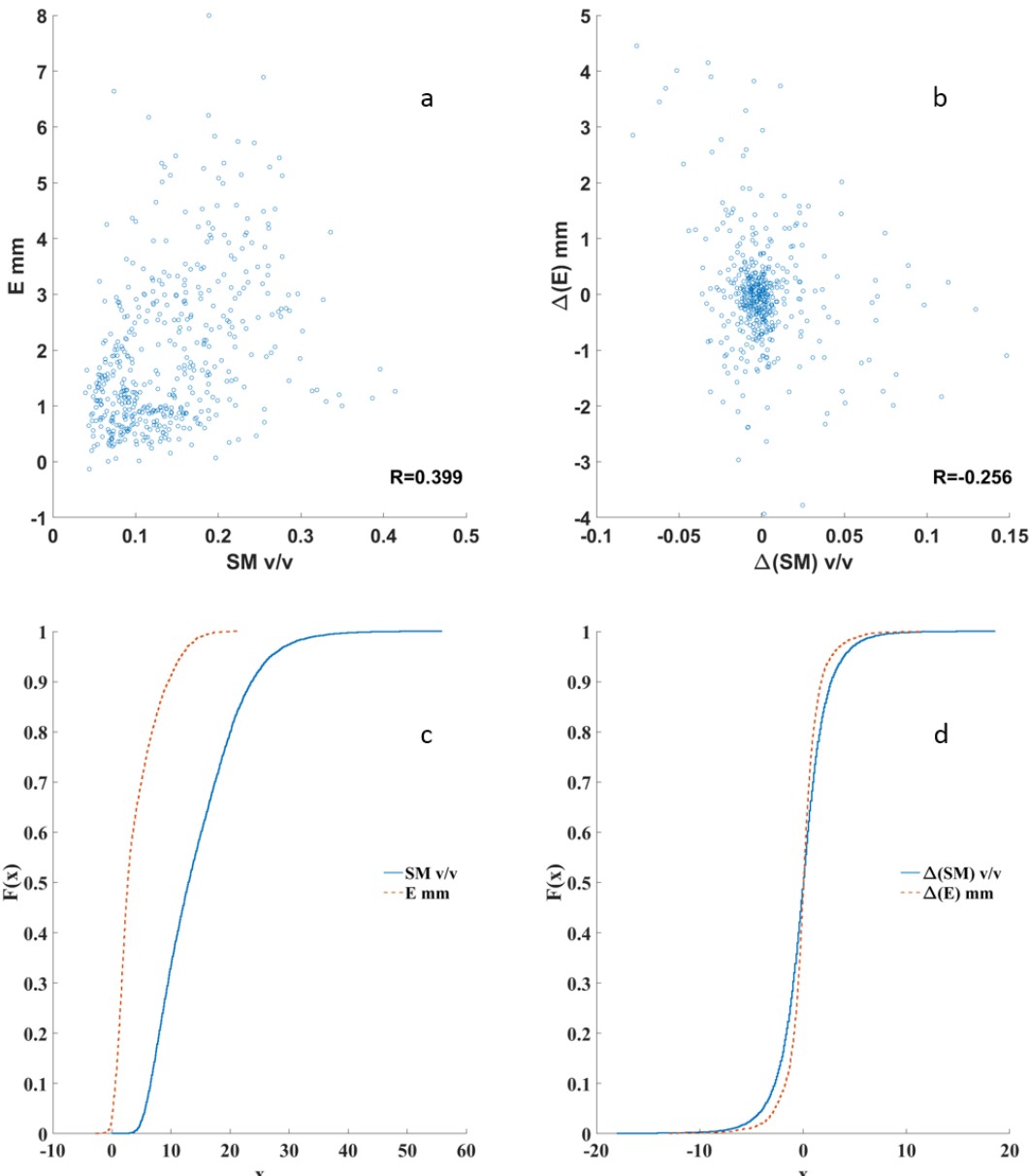

**Figure 2: (a) Scatter plot of average daily soil moisture vs. daily evaporation; (b) scatter plot of day-to-day change in average daily soil moisture vs. day-to-day change in daily evaporation; (c) cumulative distributions of average daily soil moisture and daily evaporation; (d) cumulative distributions of day-to-day change in average daily soil moisture and daily evaporation.**

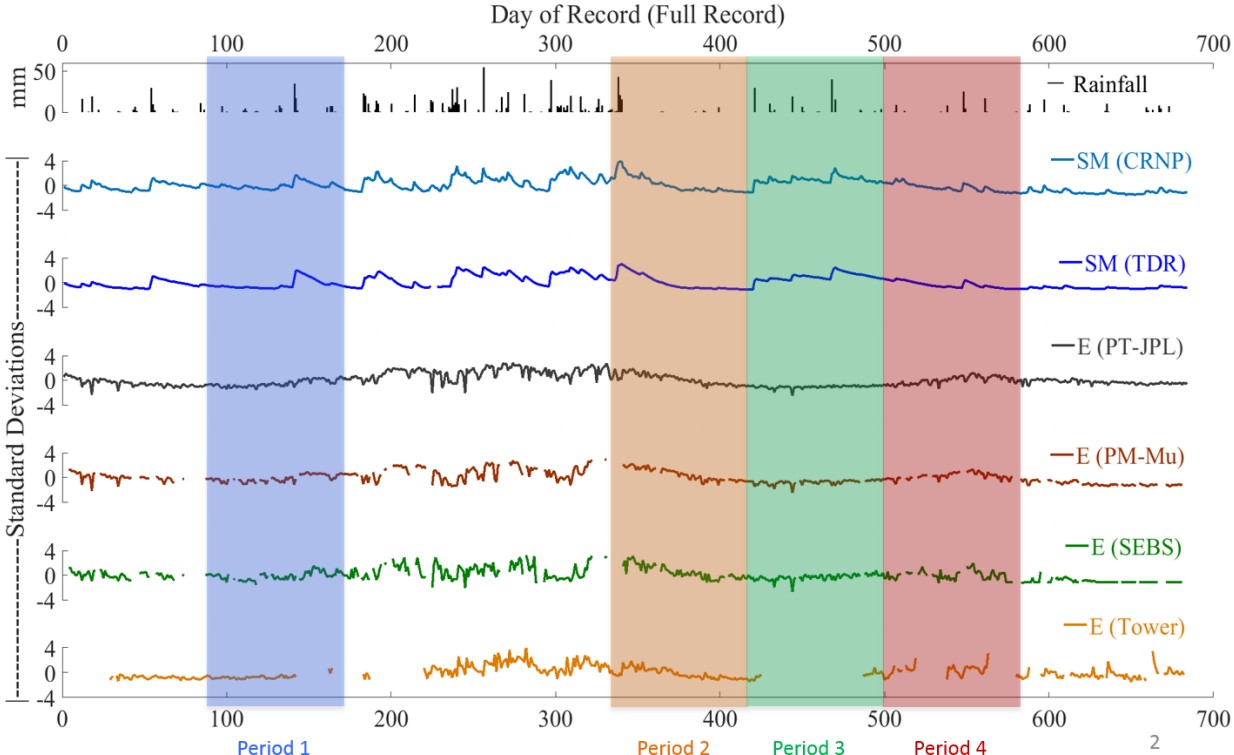

**Figure 3: Standardized soil moisture (CRNP and TDR), evaporation (modelled and observed) and precipitation signatures for entire duration of record.**

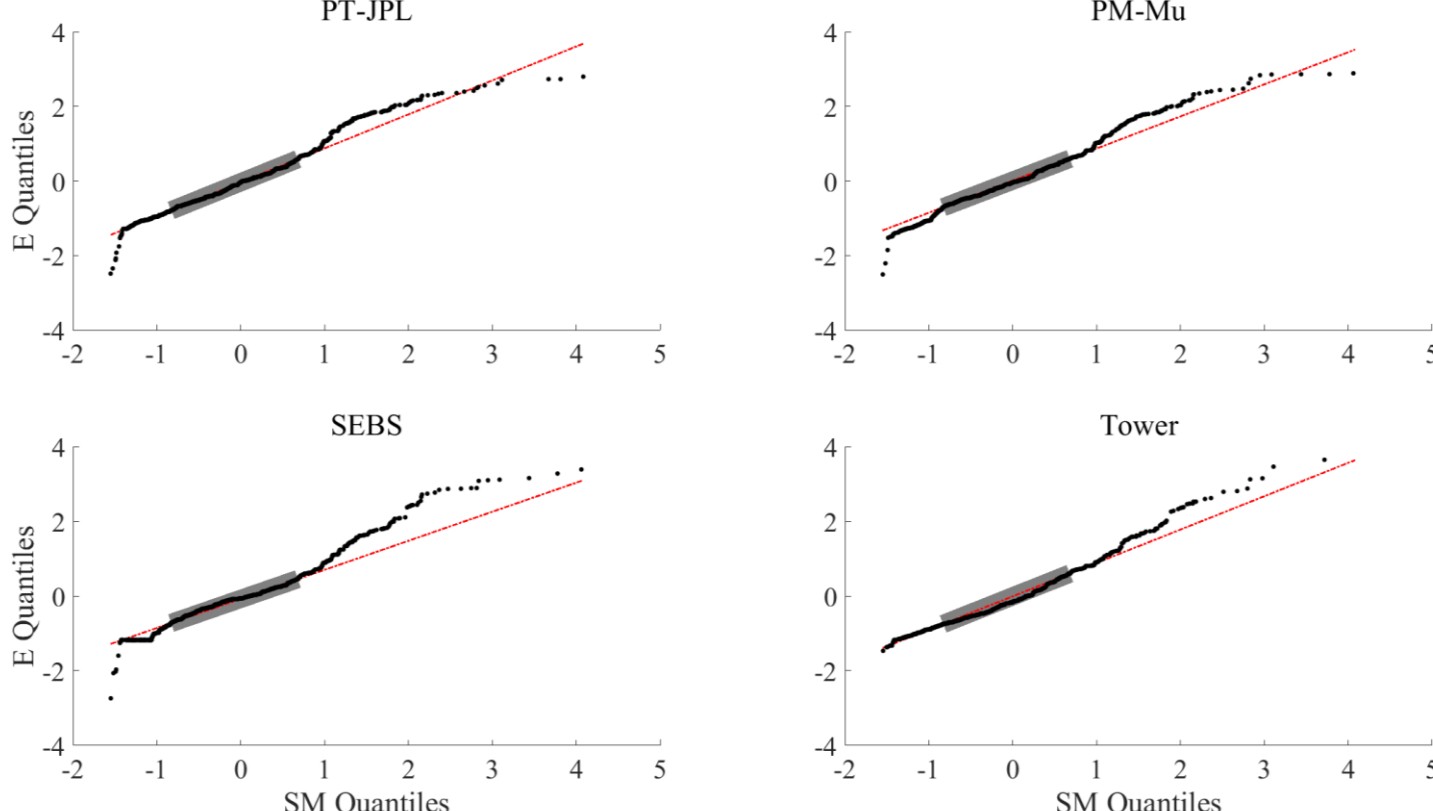

**Figure 4: Quantile-Quantile (Q-Q) plots of standardized CRNP soil moisture vs. standardized evaporation estimates and observations (at EC tower) for entire period of record.**

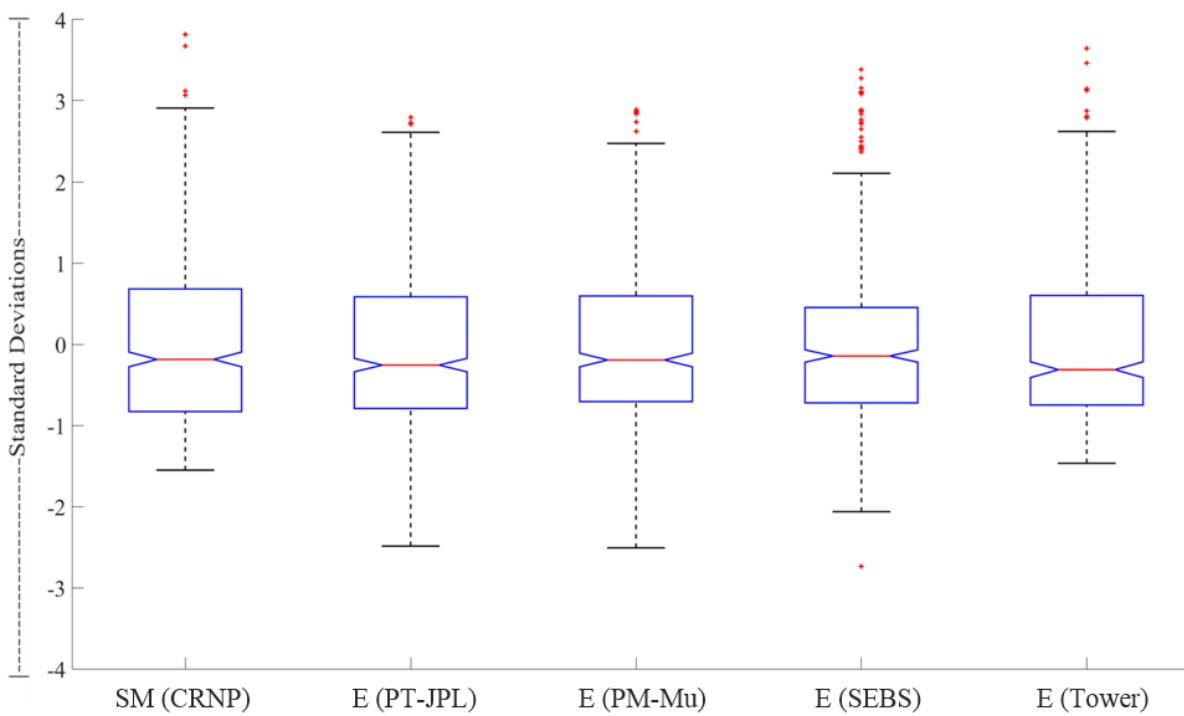

**Figure 5: ANOVA boxplots of standardized CRNP soil moisture, and standardized evaporation estimates and observations for the entire time series.**

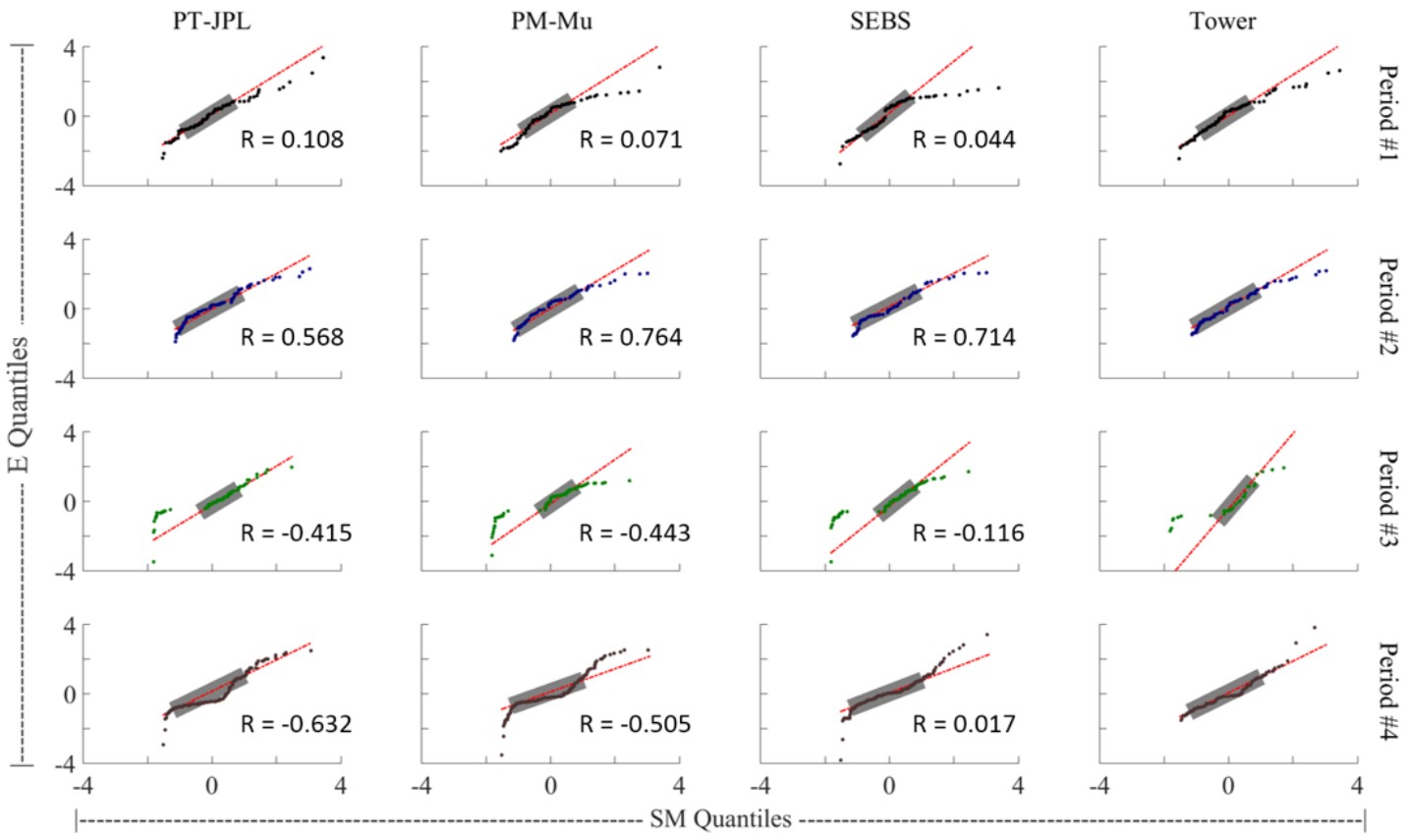

**Figure 6: Q-Q plots of standardized soil moisture vs. standardized evaporation for each defined period of analysis. R-values denote Pearson's correlation between standardized soil moisture and modelled evaporation for that particular period.**

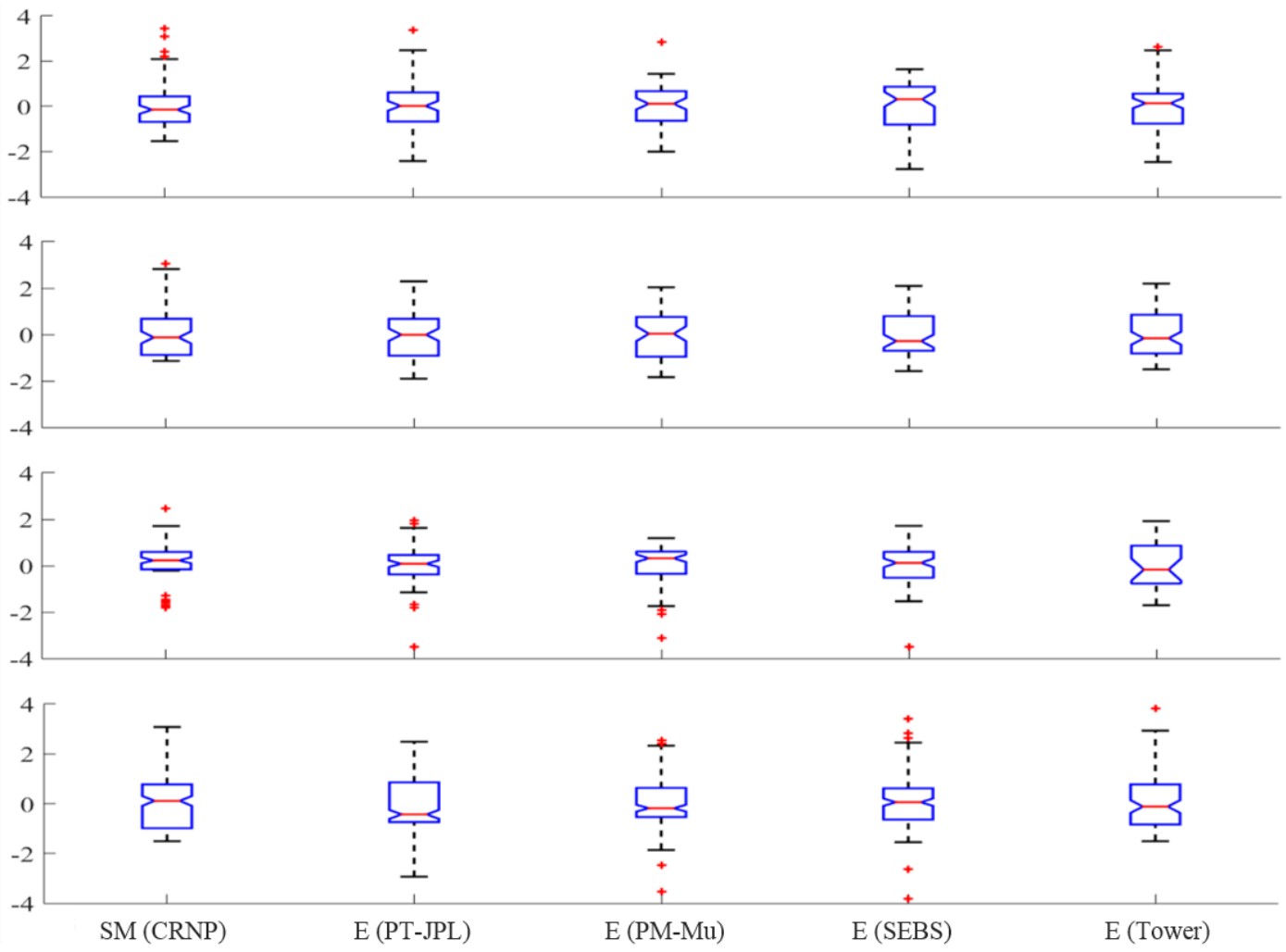

**Figure 7: ANOVA boxplots of standardized CRNP soil moisture, and standardized evaporation estimates and observations for each defined period of analysis.**

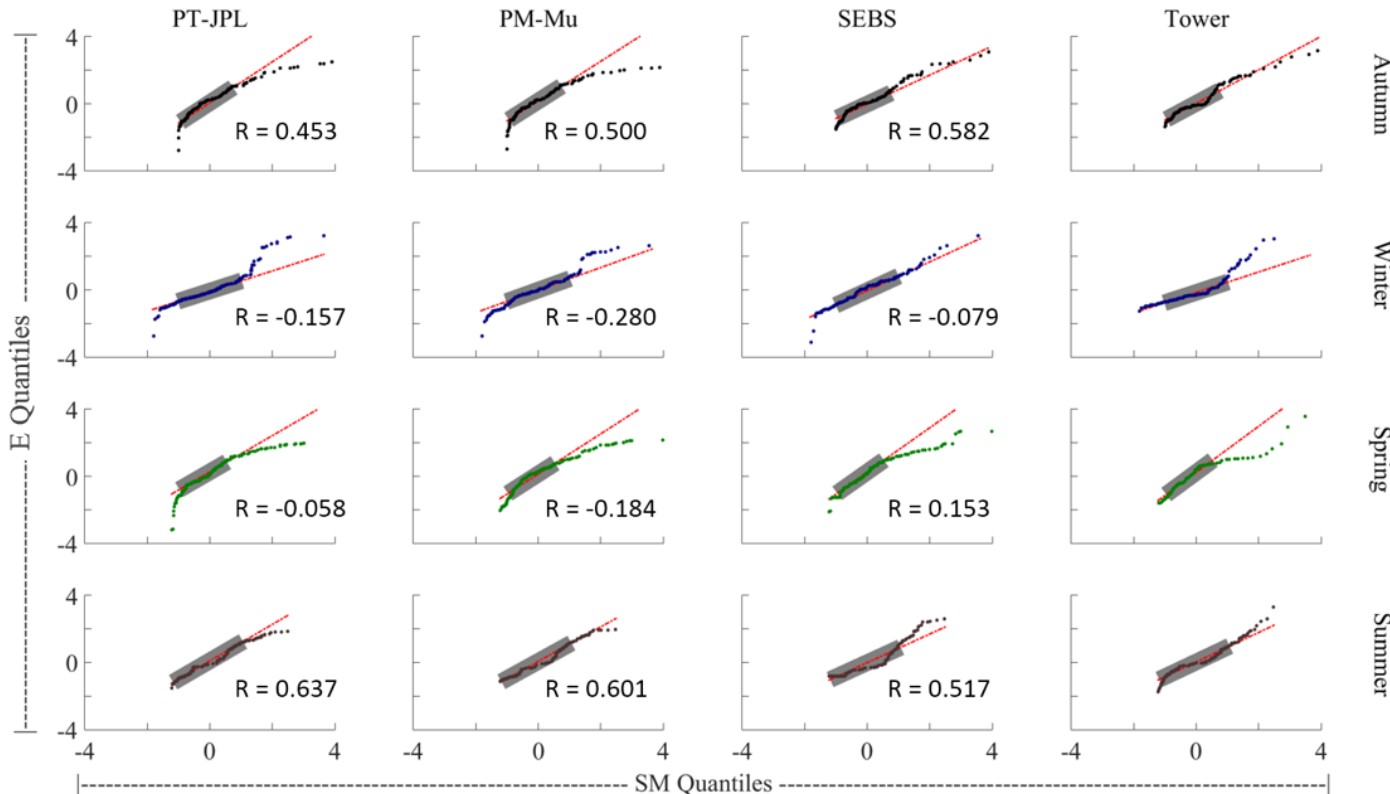

**Figure 8: Q-Q plots of standardized soil moisture vs. standardized evaporation for each season. R-values denote Pearson's correlation between standardized soil moisture and modelled evaporation for that particular season.**