# Peer review of "Examining the relationship between intermediate scale soil moisture and terrestrial evaporation within a semi-arid grassland"

_Hydrology and Earth System Sciences, 2016_

## Referee Comment (RC1) · H. Bogena (Referee) · 6 Jun 2016

This MS compares measured and modelled actual evapotranspiration (ETa) fluxes with soil moisture dynamics determined by a cosmic-ray probe for the same site located in Australia to analyse the coupling of these processes.

The MS presents an interesting application of Q–Q plots for comparing the shapes of distributions of soil moisture data and modelled fluxes of actual ET in order to evaluate coupling of land surface processes. The MS is well written and the topic fits well to the scope of this journal.

However, there are several issues regarding the methods and the interpretations of the

results (see specific comments). At this stage the results are not sufficient enough to support the interpretations and conclusions. The authors seem to have limited knowledge concerning soil hydrological processes and the CRP method and it would be advisable to add an expert of these topics to the authorship. Additional analysis of the data is needed to support the conclusions.

Chapter specific comments

1) Introduction

The introduction chapter is somewhat confused and includes several repetitions. It needs to be rewritten in a more concise and better structured way. In addition, more appropriate research questions or hypotheses need to be formulated and the structure of the paper should be presented.

There are many different terms related to processes of evaporation are used in the MS with different meanings, which is confusing for the reader. For instance, it should be stated clearly when the process of "total actual evapotranspiration" is meant, e.g. indicated with the acronym "ETa".

Instead of using the acronym "COSMOS", which is basically the US network of cosmic-ray neutron probes, the term "cosmic-ray neutron probe" or CRNP is more appropriate (see e.g. Bogena et al., 2015).

It is wrongly stated that the CRNP have footprint of 300-400 m radius and that the footprint of flux measurements by an EC-tower would be much smaller. In fact the footprint size of a CRNP typically smaller than 300 m radius (see Köhli et al., 2015) and the average footprint of an EC-tower is typically larger, integrating areas larger than 50 ha (e.g. Graf et al., 2014).

It is wrongly stated that a large number of point measurements are not feasible. However, recently established critical zone and terrestrial observatories provide exactly this kind of data (see e.g. Bogena et al., 2015; Qu et al., 2015)

2) Data and Methodology

The three models are only described very rudimentary. The basic equations and flowcharts of the algorithms should be presented to better demonstrate the differences in the methods. This information could be added as a chapter "supplementary materials". In addition, the input data used for each method should be presented separately. For instance it would be very important to know which soil moisture data was used for the modelling.

It is unclear for which reasons the TDR measurements are used in this study.

3) Analysis and Discussion

Comparing the change in root zone soil moisture with changes in ETa on a daily time scale is not appropriate, given the large differences in temporal dynamics, i.e. soil moisture changes much slower and with time lags compared to ETa, which responses to short-term changes of the meteorological forces.

Arguing that CRP and EC measurements are "rather inferred than measured" is not appropriate. To argue that these measurements a less accurate than model results is a strong statement and needs quantitative proof. Please provide measures for the accuracy of both measurements as well as for the model results.

It is argued that the CRP shows higher variability compared to TDR because it integrates over greater penetration depth. This is wrong for several reasons. First, the integral measurement of soil moisture over a profile should be less dynamic than a point measurement near the surface (e.g. 10 cm). Second, the CRP shows more dynamics, because the measurement sensitivity decreases exponentially with depth. That means the variations of the first cm below the surface are most important. In addition, the CRP is also sensitive to water stored above the surface, e.g. intercepted by leaves and litter layer (see e.g. Bogena et al., 2013).

It needs to be checked if the data standardisation has an effect on the Q–Q plots. I

might be possible that the agreement is partially due to this procedure.

I suggest to add an ANOVA test using the non-standardized data including the p-value.

Why does the SEBS model produce more outliers?

The reasoning behind the selection of the subperiods is not well visible in the data presented in Figure 3. Why is the highly dynamic and thus interesting period between subperiods 1 and 2 not included?

I have difficulties with the statement the similar distribution as shown by the Q-Q-plots alone demonstrate that ETa is driven by rot zone soil moisture. The low correlation of the raw data is telling us a different story. Therefore, this statement needs to be substantiated with further analysis.

The statement that low temperatures have decoupled soil moisture and air humidity duing period during period 4 needs to be better explained.

It is argued that long periods with no rainfall lead to a disconnection of soil moisture and ETa due to non-monotonic variations in soil moisture. I cannot follow this reasoning. Please explain in greater detail.

A soil moisture profile does not become heterogeneous. Do you mean that soil moisture gradients increase?

The statement that ETa models should be validated using soil moisture data is absurd since soil moisture is an important variable of ETa models.

Literature

Bogena, H.R., R. Bol, N. Borchard, et al. (2015): A terrestrial observatory approach for the integrated investigation of the effects of deforestation on water, energy, and matter fluxes. Science China: Earth Sciences 58(1): 61-75, doi: 10.1007/s11430-014-4911-7.

Bogena, H.R., J.A. Huisman, C. Hübner, J. Kusche, F. Jonard, S.Vey, A. Güntner and

H. Vereecken (2015): Emerging methods for non-invasive sensing of soil moisture dynamics from field to catchment scale: A review. WIREs Water 2(6): 635–647, doi: 10.1002/wat2.1097.

Bogena, H.R., J.A. Huisman, R. Baatz, R., H.-J. Hendricks Franssen and H. Vereecken (2013): Accuracy of the cosmic-ray soil water content probe in humid forest ecosystems: The worst case scenario. Water Resour. Res. 49 (9): 5778-5791, doi: 10.1002/wrcr.20463.

Graf, A., H.R. Bogena, C. Drüe, H. Hardelauf, T. Pütz, G. Heinemann and H. Vereecken (2014). Spatiotemporal relations between water budget components and soil water content in a forested tributary catchment. Water Resour. Res. 50(6): 4837–4857, doi: 10.1002/2013WR014516.

Köhli, M., Schrön, M., Zreda, M., Schmidt, U., Dietrich, P. and Zacharias, S.: Footprint characteristics revised for field-scale soil moisture monitoring with cosmic-ray neutrons. Water Resour. Res., 2015.

Qu, W., H.R. Bogena., J.A. Huisman, J. Vanderborght, M. Schuh, E. Priesack and H. Vereecken (2015): Predicting sub-grid variability of soil water content from basic soil information. Geophys. Res.Lett. 42: 789–796, doi:10.1002/2014GL062496.

---

## Referee Comment (RC2) · Anonymous Referee #2 · 9 Jun 2016

The authors present an interesting case study comparing three different commonly used evaporation schemes versus a COSMOS soil moisture probe. The results illustrate reasonable statistical comparisons between the methods between the 25th and 75th quantile, but breakdown outside these ranges. I agree with the authors assessment of the challenges comparing the state variable of soil moisture with evaporation flux, particularly given the spatial scale differences of the observations. The work here is a valuable contribution to continue advancing the utility of the COSMOS soil moisture probes with applications in surface energy balance or land atmospheric coupling. The paper is well written and suitable for HESS. Below are some recommendations to improve the manuscript.

Comments:

Pg 2. L2. Is it land surface evaporation or evapotranspiration? The symbol ET is a bit confusing if it only refers to evaporation only.

Pp 6. L11-19. Is the COSMOS data the same as presented by Hawdon 2014? That is, it is corrected for water vapor, geomagnetic latitude, pressure in the same way? Please specify.

P 8 L24. The selection of sampling periods seems a bit arbitrary. Why not use seasons or PET to separate periods?

L 10 L31. I am not what is might by this sentence, the soil moisture profile becomes heterogeneous during periods when it is disconnected to the atmosphere? Can you please explain more or show an example?

Pg 11 L13 and Figure 2a. The comparison between soil moisture and ET should be further partitioned by PET amount or season. Following the simple broken stick type model in Rodriguez-Iturbe 2001 and Laio 2001, I would expect there to be a family of curves with the plateau being near ETmax for each set of curves. I suggest the authors organize the data by season or PET groups and replot (with either colors or different symbols). For such a simple dryland grassland site I would expect the broken stick kind of model to represent this data well. The direct correspondence between soil moisture and ET may become more clear instead of just the distributions. If so things like the soil moisture threshold at which ET is reduced may become clear from the datasets.

Rodriguez-Iturbe, I., A. Porporato, F. Laio, and L. Ridolfi (2001), Plants in water-controlled ecosystems: active role in hydrologic processes and response to water stress - I. Scope and general outline, Adv. Water Resour., 24(7), 695-705.

Laio, F., A. Porporato, L. Ridolfi, and I. Rodriguez-Iturbe (2001), Plants in water-controlled ecosystems: active role in hydrologic processes and response to water stress - II. Probabilistic soil moisture dynamics, Adv. Water Resour., 24(7), 707-723.

Comments on conclusions: The challenge of relating energy balance models like SEBS to soil moisture has some interesting applications. For example, in agriculture many research and private industry groups are using such routines from satellites and drones to schedule irrigation. However, the soil moisture may be more unconstrained in this case than can be suitable for reasonable management of irrigation amounts and timing. The authors could potentially comment on this application given the findings of the paper.

---

## Referee Comment (RC3) · H. Bogena (Referee) · 16 Jun 2016

Response to the author's response

Thanks for the response and clarifications. However, in my view, not all the responses are sufficient. Therefore, I have commented those responses which I feel were not appropriate.

The footprint area cited by the reviewer from the article by Graf et al. (2014) is based on a tower height of 38m, which is much higher than the tower at this study location.

> *What is important is the height of the EC sensors above canopy, which is only about 12 m in the case of Graf et al. (2014).*

While a limited number of observatories are providing fine resolution soil moisture data, they involve significant outlay of finance, physical effort and time, as compared to, for example, utilizing scaling schemes, or installation of intermediate scale sensors. Although such observatories are invaluable in providing data to understand the underlying processes, it remains impractical to implement a large number of sensors across any and every field of interest.

> *An increasing number of existing large scale sensor networks are make their data freely available to the science community (e.g. SCAN, ICOS). In addition, a number a measurement techniques are emerging that make use of existing networks that formally were installed for other reasons (e.g. Bogena et al., 2015) and thus will provide a much better coverage of soil moisture observation in the near future beyond the observatories.*

Given their extensive appearance in the literature, we feel that it is not necessary to repeat the description of these models in intricate detail in this paper.

> *I am still very much more in favor for adding this information. Why should the reader gather all this papers himself to get a basic overview of the models and their differences? Presenting this information makes the paper much more comprehensible and also better explains why the three models were used instead of only one.*

The ancillary TDR measurements were used to confirm the validity of the CRNP soil moisture time series.

> *Actually, the CRNP validity was not tested in this paper in a strict sense. This could not be done with a single TDR profile anyway, since a network of point measurements within the CRNP footprint would be needed to do this (see e.g. Bogena et al., 2013).*

That being said, nowhere in the manuscript do we imply or insinuate that the observations are less accurate than model results. Without particular reference, we are not sure where the reviewer gets this impression in the text.

*It is true that it was not implied that the observations were less accurate than the model results. The impression arises, because the focus was led on the comparison of soil moisture with model results.*

Since the line plots of the soil moisture and evaporation do not match, but the distributions do, it is a logical inference that the two quantities are behaving similarly at the distribution level. Hence, the deduction that the soil moisture (root zone since the CRNP is measuring over depth) is still driving the evaporation process.

*First, I have to repeat again that the term "evaporation" is confusing. I guess you are referring to evapotranspiration because it is related to root zone (i.e. evaporation from bare soil and intercepted water is not related to root water uptake in the root zone). So again, please improve the terminology in the paper.*

*What I was trying to point out is that the existence of similar distributions alone is not adequate for this deduction, because processes at the soil-vegetation-atmosphere interface tend to be very complex. For instance, the process of evaporation from canopy is completely independent from soil moisture, but it might be an important part of total evapotranspiration at this location. Therefore, the distributions of both quantities might be similar because the CRNP measurements are also influenced by water intercepted on the canopy (see e.g. Bogena et al., 2013).*

*In addition, the CRNP typically does not cover the whole root zone, because for intermediate soil moisture ranges the penetration depth is restricted to 20-30 cm. In addition, the CNRS is much more sensible to soil moisture of the first centimeters. In this sense the TDR measurements might even better represent root zone soil moisture at this site. Thus, the Q-Q analysis should also be done with the TDR data to test the assumption soil moisture is driven by the evapotranspiration process.*

This seems a case of a misunderstanding by the reviewer regarding the evaporation models, rather than an absurdity on the part of the authors. In the range of models examined, and in the vast majority of satellite based evaporation models, soil moisture does not feature as an input variable.

*I am sorry for my ignorance concerning the models used in this study. Clearly, a better description of the models will help the readers to be better following the reasoning presented in the paper. If soil moisture is not a model variable this should be explicitly mentioned in the paper. Otherwise the modelled soil moisture should be compared with the measured soil moisture to demonstrate the validity of the model.*

---

## Author Comment (AC1) · 16 Jun 2016

**Response to comments by Referee #1 (H. Bogena) on "Examining the relationship between intermediate scale soil moisture and terrestrial evaporation within a semi-arid grassland" by Jana et al.**

> *We greatly appreciate the review comments and thank the reviewer for their effort. We have addressed all of the comments and present our responses below.*
>
> *The review comments are in regular typeface, while all responses are in italics and indented paragraphs.*

This MS compares measured and modelled actual evapotranspiration (ETa) fluxes with soil moisture dynamics determined by a cosmic-ray probe for the same site located in Australia to analyse the coupling of these processes.

The MS presents an interesting application of Q–Q plots for comparing the shapes of distributions of soil moisture data and modelled fluxes of actual ET in order to evaluate coupling of land surface processes. The MS is well written and the topic fits well to the scope of this journal.

However, there are several issues regarding the methods and the interpretations of the results (see specific comments). At this stage the results are not sufficient enough to support the interpretations and conclusions. The authors seem to have limited knowledge concerning soil hydrological processes and the CRP method and it would be advisable to add an expert of these topics to the authorship. Additional analysis of the data is needed to support the conclusions.

> *We thank the reviewer for their supportive comments regarding the manuscript. However, we disagree with the reviewer's thoughts on apparent issues related to the methods and interpretation of our results. We consider the results as sound and suspect there has been some misunderstanding of the approach and the analysis undertaken. We have endeavored to make this much clearer in the revised manuscript and in the responses to the specific comments below.*

Chapter specific comments

1) Introduction
The introduction chapter is somewhat confused and includes several repetitions. It needs to be rewritten in a more concise and better structured way. In addition, more appropriate research questions or hypotheses need to be formulated and the structure of the paper should be presented.

> *We have gone through this section with an eye towards readability and comprehension and have rearranged sections as well as more clearly articulating the rationale and motivation behind this work. We have stated the motivation (P3, L6-9, "… identifying complimentary observation sources that can be used to improve upon the evaluation of a variety of hydrological processes is a much needed objective (McCabe et al. 2008). This critical need forms a key motivation of this work where we look for an answer to the question: are independent hydrological data-sets available that can be used to inform upon linked elements of the hydrological cycle?"), the rationale (P4, L6-10, "With improved sensing of the root-zone soil moisture, it is expected that any modelled relationship between evaporation and soil moisture will be more robust. From an observational standpoint, however, it has been challenging to explore these links directly due to the mismatch in data scales. Using CRNP soil moisture data collocated with gridded model estimates of evaporation may provide some insight into these processes and relationships.") and the objective (P4, L12-13, "… the objective of this study is to investigate the potential of using CRNP soil moisture retrievals to*

*evaluate modelled evaporation estimates derived from a combination of tower based and remote sensing inputs.").*

There are many different terms related to processes of evaporation are used in the MS with different meanings, which is confusing for the reader. For instance, it should be stated clearly when the process of "total actual evapotranspiration" is meant, e.g. indicated with the acronym "ETa".

*We have updated the manuscript and only use the term "evaporation" to represent land surface evaporation, which comprises of evaporation from soil and canopy, as well as transpiration from vegetation. This is standard and accepted terminology.*

Instead of using the acronym "COSMOS", which is basically the US network of cosmicray neutron probes, the term "cosmic-ray neutron probe" or CRNP is more appropriate (see e.g. Bogena et al., 2015).

*We have replaced "COSMOS" with "CRNP" in all cases.*

It is wrongly stated that the CRNP have footprint of 300-400 m radius and that the footprint of flux measurements by an EC-tower would be much smaller. In fact the footprint size of a CRNP typically smaller than 300 m radius (see Köhli et al., 2015) and the average footprint of an EC-tower is typically larger, integrating areas larger than 50 ha (e.g. Graf et al., 2014).

*We have revised the text to reflect the updated footprint radius of the CRNP sensor, as given by Kohli et al, 2015.*

*The dynamic footprint of an EC tower depends on the height of the tower and roughness of the surface, as well as wind-speed influences. For a 2 m tower height and pasture, the footprint remains within the range of a few hundred meters, with the standard rule of 1:100 for fetch distance as validated by Leclerc and Thurtell (1990). We have updated the text (P5, L31) to specify the height of the EC tower. In our particular case (and a large number of Fluxnet installations), the fetch of the EC tower is comparable to the scale of the CRNP footprint.*

*The footprint area cited by the reviewer from the article by Graf et al. (2014) is based on a tower height of 38m, which is much higher than the tower at this study location.*

*Leclerc, M.Y., & Thurtell, G.W. (1990). Footprint prediction of scalar fluxes using a Markovian analysis. Boundary-Layer Meteorology, 52, 247-258*

It is wrongly stated that a large number of point measurements are not feasible. However, recently established critical zone and terrestrial observatories provide exactly this kind of data (see e.g. Bogena et al., 2015; Qu et al., 2015)

*While a limited number of observatories are providing fine resolution soil moisture data, they involve significant outlay of finance, physical effort and time, as compared to, for example, utilizing scaling schemes, or installation of intermediate scale sensors. Although such observatories are invaluable in providing data to understand the underlying processes, it remains impractical to implement a large number of sensors across any and every field of interest. The statement remains very much correct in this context, but we have adjusted the text to include the observatory concept.*

2) Data and Methodology
The three models are only described very rudimentary. The basic equations and flowcharts of the algorithms should be presented to better demonstrate the differences in the methods. This information could be added as a chapter "supplementary materials".

> *The three models used in this study are well-established and commonly used models, with extensive coverage in the recent literature. We have provided a number of key references to articles in which they are described in more detail. We also provide references to articles where the three methods are compared to each other. Given their extensive appearance in the literature, we feel that it is not necessary to repeat the description of these models in intricate detail in this paper.*

In addition, the input data used for each method should be presented separately. For instance it would be very important to know which soil moisture data was used for the modelling. It is unclear for which reasons the TDR measurements are used in this study.

> *There seems to be some misunderstanding in relation to the evaporation models used here. No soil moisture observations were used in any of the three models mentioned in this study, as it is not a requirement. To make this clearer to the reader, we have mentioned the inputs required for each in the model descriptions.*

> *The ancillary TDR measurements were used to confirm the validity of the CRNP soil moisture time series.*

3) Analysis and Discussion
Comparing the change in root zone soil moisture with changes in ETa on a daily time scale is not appropriate, given the large differences in temporal dynamics, i.e. soil moisture changes much slower and with time lags compared to ETa, which responses to short-term changes of the meteorological forces.

> *Fundamentally, changes in soil moisture will ultimately be reflected in changes in the evaporation response: the issue of time scale is of course critical in this mass balance approach. For this setting, examining changes at the daily scale seems to be an appropriate resolution, as borne out by the results. It is precisely because of this temporal mismatch that the quantities are compared at the distribution level, rather than point to point. As expressed in Figure 2, the day to day changes in the soil moisture and ET follow very similar distributions.*

Arguing that CRP and EC measurements are "rather inferred than measured" is not appropriate. To argue that these measurements a less accurate than model results is a strong statement and needs quantitative proof. Please provide measures for the accuracy of both measurements as well as for the model results.

> *The reviewer appears to have misunderstood the statement in P7, L20. We mention that the observations of soil moisture and evaporation are "inferred" as in they are indirectly measured. For example, field sensors do not directly measure the soil moisture, but other quantities such as the dielectric constant or the neutron counts. These are then converted to the quantity of interest, i.e., the soil moisture, using a conversion algorithm. Uncertainties and errors are bound to be introduced at each step, thus reducing accuracy. This is an entirely appropriate (and well accepted) rationale.*

> *That being said, nowhere in the manuscript do we imply or insinuate that the observations are less accurate than model results. Without particular reference, we are not sure where the reviewer gets this impression in the text.*

It is argued that the CRP shows higher variability compared to TDR because it integrates over greater penetration depth. This is wrong for several reasons. First, the integral measurement of soil moisture over a profile should be less dynamic than a point measurement near the surface (e.g. 10 cm). Second, the CRP shows more dynamics, because the measurement sensitivity decreases exponentially with depth. That means the variations of the first cm below the surface are most important. In addition, the CRP is also sensitive to water stored above the surface, e.g. intercepted by leaves and litter layer (see e.g. Bogena et al., 2013).

> *This is certainly valid reasoning and we have adjusted the text to better reflect our meaning. It should be noted that the CS616 TDR used at the site also provides a (vertically) depth integrated measurement (0-30 cm), not a truly point scale estimate. The text now reads as:*
>
> *"While it might be expected that the TDR data should display greater variability in response, the CRNP measurements have higher variability in soil moisture values. This could be due to factors such as the variability in the measurement depth of the CRNP with change in the saturation, and higher sensitivity of the CRNP to near-surface moisture as compared to deeper layers (Bogena et al. 2013). High frequency variations at the soil surface may also be attenuated in the TDR signal since it is integrated over the 30 cm probe depth."*
>
> *Regarding the sensitivity of the measurement, the soil moisture data has been corrected for other sources as explained by Hawdon et al., 2014. At this semi-arid rangeland site, interception and litter are not factors of great influence.*

It needs to be checked if the data standardisation has an effect on the Q–Q plots. I might be possible that the agreement is partially due to this procedure. I suggest to add an ANOVA test using the non-standardized data including the p-value.

> *Standardizing the data has no effect on the shape of the Q-Q plots, since the plots depend on the residuals. The only difference is in the numerical value of the axes. An example is shown below with CRNP soil moisture and PT-JPL evaporation data from the entire period of record. The first panel is the Q-Q plot for the raw data, while the lower panel is for standardized data. As can be seen, there is no difference in the shape of the plot. Standardizing the data has just scaled the values differently.*

[Figure]

*As mentioned in the text (P8, L16-17), the p-value is an indicator of the similarity of the mean values of the two quantities being compared. In our case, the two quantities, soil moisture and evaporation, have vastly different ranges and units, and thus, mean values. Performing an ANOVA test between two such datasets can only provide a p-value of 0, as shown in the example below. The ANOVA is performed on the raw dataset for the entire period of record used in the study. As can be seen, the boxplot for the soil moisture is crushed due to the range of the evaporation data, and the p-value is essentially 0.*

[Figure]

Why does the SEBS model produce more outliers?

> *The SEBS model is more sensitive to uncertainties in land surface temperature as compared to the other two models. This could make the model outputs behave in a different manner, especially at the extremes. (P11, L19-21)*

The reasoning behind the selection of the subperiods is not well visible in the data presented in Figure 3. Why is the highly dynamic and thus interesting period between subperiods 1 and 2 not included?

> *The period between sub-periods 1 and 2, while highly dynamic, was close to being simply a scaled version of the total period of record. This is borne out in the Q-Q plots below for that period (DoR 230-320).*

[Figure]

> *Hence, it was felt that analyzing periods of distinctly different behaviors, as described in the text for the four chosen sub-periods, would provide a better understanding of the correspondence between the soil moisture and evaporation signals under different situations.*

I have difficulties with the statement the similar distribution as shown by the Q-Q-plots alone demonstrate that ETa is driven by rot zone soil moisture. The low correlation of the raw data is telling us a different story. Therefore, this statement needs to be substantiated with further analysis.

> *Q-Q plots are a commonly used tool to assess similarity of distributions. Correlation is a point-to-point statistic. We have shown in the manuscript (and referred to other studies with similar deductions) that such point-to-point statistics are not necessarily the best way to evaluate correspondence between two stochastic variables, especially those whose process time scales are different. In such a scenario, distribution matching is a better option. Since the line plots of the soil moisture and evaporation do not match, but the distributions do, it is a logical inference that the two quantities are behaving similarly at the distribution level. Hence, the deduction that the soil*

*moisture (root zone since the CRNP is measuring over depth) is still driving the evaporation process.*

The statement that low temperatures have decoupled soil moisture and air humidity duing period during period 4 needs to be better explained.

> *We have updated the text with additional explanation. The section now reads as:*
>
> *"It is also likely that low temperatures and additional hydro-meteorological factors could have caused a de-coupling of the soil moisture from the air humidity. Due to the low temperatures, the air humidity would be lower, while the frequent precipitation ensures high soil moisture content. This creates a steep gradient for the moisture at the soil-air interface. In such a scenario, regardless of the presence of abundant soil moisture for evaporation, the models which use air humidity as a surrogate for soil moisture may report lower estimates compared to those observed from the eddy-covariance tower."*

It is argued that long periods with no rainfall lead to a disconnection of soil moisture and ETa due to non-monotonic variations in soil moisture. I cannot follow this reasoning. Please explain in greater detail.
A soil moisture profile does not become heterogeneous. Do you mean that soil moisture gradients increase?

> *The soil moisture profile is said to become heterogeneous since the surface and deeper layer moistures are driven by different processes, and are not linked to each other. We have updated the text with some additional explanation for the terminology. The section now reads as:*
>
> *"However, there were also long periods with no rainfall events. Combined with the higher temperatures of summer, this leads to greater non-monotonic variations in the soil moisture signature, thus creating a disconnect with the evaporation patterns. There are more switches between moisture-constrained and energy-constrained conditions during this season. It has been demonstrated previously that the occurrence of hot and dry periods leads to de-coupling of soil moisture and evaporation (Pollacco and Mohanty 2012). The soil moisture profile in such situations becomes heterogeneous in that the process driving the surface soil moisture variability (mainly soil evaporation) no longer influences the deeper layer soil moisture variability (mainly due to transpiration). Further explanation of this de-coupling process can be found in the article by Pollacco and Mohanty (2012)."*

The statement that ETa models should be validated using soil moisture data is absurd since soil moisture is an important variable of ETa models.

> *This seems a case of a misunderstanding by the reviewer regarding the evaporation models, rather than an absurdity on the part of the authors. In the range of models examined, and in the vast majority of satellite based evaporation models, soil moisture does not feature as an input variable. Here we exploit the physical mechanism that makes soil moisture a key driver of the evaporation process. As such, it makes perfect sense to validate models using observations that govern or influence that process to a significant extent. It is the same rationale that one might use to evaluate spatially distributed soil moisture maps by using rainfall fields. Indeed, it is this reasoning that is at the heart of the approach explored here. Given the general lack of observation data concerning any specific process, it is important that independently observed, yet physically linked variables, be used to aid in the evaluation process.*

Literature

Bogena, H.R., R. Bol, N. Borchard, et al. (2015): A terrestrial observatory approach for the integrated investigation of the effects of deforestation on water, energy, and matter fluxes. Science China: Earth Sciences 58(1): 61-75, doi: 10.1007/s11430-014-4911-7.

Bogena, H.R., J.A. Huisman, C. Hübner, J. Kusche, F. Jonard, S.Vey, A. Güntner and H. Vereecken (2015): Emerging methods for non-invasive sensing of soil moisture dynamics from field to catchment scale: A review. WIREs Water 2(6): 635–647, doi: 10.1002/wat2.1097.

Bogena, H.R., J.A. Huisman, R. Baatz, R., H.-J. Hendricks Franssen and H. Vereecken (2013): Accuracy of the cosmic-ray soil water content probe in humid forest ecosystems: The worst case scenario. Water Resour. Res. 49 (9): 5778-5791, doi: 10.1002/wrcr.20463.

Graf, A., H.R. Bogena, C. Drüe, H. Hardelauf, T. Pütz, G. Heinemann and H. Vereecken (2014). Spatiotemporal relations between water budget components and soil water content in a forested tributary catchment. Water Resour. Res. 50(6): 4837–4857, doi: 10.1002/2013WR014516.

Köhli, M., Schrön, M., Zreda, M., Schmidt, U., Dietrich, P. and Zacharias, S.: Footprint characteristics revised for field-scale soil moisture monitoring with cosmic-ray neutrons. Water Resour. Res., 2015.

Qu, W., H.R. Bogena., J.A. Huisman, J. Vanderborght, M. Schuh, E. Priesack and H. Vereecken (2015): Predicting sub-grid variability of soil water content from basic soil information. Geophys. Res.Lett. 42: 789–796, doi:10.1002/2014GL062496.

---

## Author Comment (AC2) · 16 Jun 2016

**Response to comments by Referee #2 (anonymous) on "Examining the relationship between intermediate scale soil moisture and terrestrial evaporation within a semi-arid grassland" by Jana et al.**

> *We greatly appreciate the review comments and thank the reviewer for their effort. We have addressed all of the comments and present our responses below.*
>
> *The review comments are in regular typeface, while all responses are in italics and indented paragraphs.*

The authors present an interesting case study comparing three different commonly used evaporation schemes versus a COSMOS soil moisture probe. The results illustrate reasonable statistical comparisons between the methods between the 25th and 75th quantile, but breakdown outside these ranges. I agree with the authors assessment of the challenges comparing the state variable of soil moisture with evaporation flux, particularly given the spatial scale differences of the observations. The work here is a valuable contribution to continue advancing the utility of the COSMOS soil moisture probes with applications in surface energy balance or land atmospheric coupling.

The paper is well written and suitable for HESS. Below are some recommendations to improve the manuscript.

> *We thank the anonymous reviewer for their positive comments on the manuscript.*

Comments:
Pg 2. L2. Is it land surface evaporation or evapotranspiration? The symbol ET is a bit confusing if it only refers to evaporation only.

> *We have removed all instances of the term "ET" in the manuscript. Following the convention of (Kalma et al. 2008) the term "evaporation" is used to represent land surface evaporation, which comprises evaporation from soil and canopy, as well as transpiration from vegetation.*
>
> *Kalma, J.D., McVicar, T.R., & McCabe, M.F. (2008). Estimating land surface evaporation: A review of methods using remotely sensed surface temperture data. Surveys in Geophysics, 29, 421-469*

Pp 6. L11-19. Is the COSMOS data the same as presented by Hawdon 2014? That is, it is corrected for water vapor, geomagnetic latitude, pressure in the same way? Please specify.

> *Yes, the data is the same as presented by Hawdon et al., 2014. We have updated the text to clarify this point.*

P 8 L24. The selection of sampling periods seems a bit arbitrary. Why not use seasons or PET to separate periods?

> *We agree that the sampling periods are somewhat arbitrary. However, we felt that analyzing periods of distinctly different hydrological behavior, as described in the text for the four chosen sub-periods, would provide a better understanding of the correspondence between the soil moisture and evaporation signals under different situations. In regards to the reviewer's suggestion on a*

*more formal allocation of analysis periods, we analyzed the relationship based on partitioning the time series according to seasons. As stated in the text, the results were similar to those obtained by partitioning by behavior in that while the SEBS model performed better than the others, no single model output corresponded well with the soil moisture across all seasons.*

*The PET for the site (computed using meteorological tower data) follows a distinctly seasonal trend (see figure below). As such, partitioning based on seasons can be deemed analogous to using PET.*

[Figure]

L 10 L31. I am not what is might by this sentence, the soil moisture profile becomes heterogeneous during periods when it is disconnected to the atmosphere? Can you please explain more or show an example?

*The soil moisture profile is said to become heterogeneous since the surface and deeper layer moistures are driven by different processes, and are not linked to each other. We have updated the text with some more explanation for the term. The section now reads thus:*

*"However, there were also long periods with no rainfall events. Combined with the higher temperatures of summer, this leads to greater non-monotonic variations in the soil moisture signature, thus creating a disconnect with the evaporation patterns. There are more switches between moisture-constrained and energy-constrained conditions during this season. It has been demonstrated previously that the occurrence of hot and dry periods leads to de-coupling of soil moisture and evaporation (Pollacco and Mohanty 2012). The soil moisture profile in such situations becomes heterogeneous in that the process driving the surface soil moisture variability (mainly soil evaporation) no longer influences the deeper layer soil moisture variability (mainly due to transpiration). Further explanation of this de-coupling process can be found in Pollacco and Mohanty (2012)."*

Pg 11 L13 and Figure 2a. The comparison between soil moisture and ET should be further partitioned by PET amount or season.

*We did plot the scatter of soil moisture v. evaporation after partitioning by seasons (please see figure below). However, it was felt that such a comparison did not add much new information apart*

*from showing that the relationships between the two quantities were different in different seasons. Since we already present this difference in the other plots where we analyze the relationship based on seasons, we decided not to include this figure and comparison in the manuscript.*

[Figure]

Following the simple broken stick type model in Rodriguez-Iturbe 2001 and Laio 2001, I would expect there to be a family of curves with the plateau being near ETmax for each set of curves. I suggest the authors organize the data by season or PET groups and replot (with either colors or different symbols). For such a simple dryland grassland site I would expect the broken stick kind of model to represent this data well. The direct correspondence between soil moisture and ET may become more clear instead of just the distributions. If so things like the soil moisture threshold at which ET is reduced may become clear from the datasets.

*This is an excellent suggestion, and we thank the reviewer for it. Based on this suggestion, we performed piece-wise linear regression analyses on the seasonally-partitioned data. The plots are given below:*

[Figure]

*Each row in the figure corresponds to a model (PT-JPL, PM-Mu, and SEBS), while each column corresponds to a season (autumn, winter, spring, summer). Unfortunately, in this particular case there was not much useful information that could be gleaned from the plots. Some information regarding the soil moisture threshold where the evaporation rate starts decreasing can be discerned for all three models only in the summer. The plots for the other seasons are largely non-informative. However, it should be borne in mind that this data spans a relatively short period (under 2 years). It is possible that a longer data record over many more years could result in a more distinct behavior as expected by the reviewer. For the present study, we decided against including this analysis in the manuscript.*

Rodriguez-Iturbe, I., A. Porporato, F. Laio, and L. Ridolfi (2001), Plants in watercontrolled ecosystems: active role in hydrologic processes and response to water stress - I. Scope and general outline, Adv. Water Resour., 24(7), 695-705.

Laio, F., A. Porporato, L. Ridolfi, and I. Rodriguez-Iturbe (2001), Plants in watercontrolled ecosystems: active role in hydrologic processes and response to water stress - II. Probabilistic soil moisture dynamics, Adv. Water Resour., 24(7), 707-723.

Comments on conclusions: The challenge of relating energy balance models like SEBS to soil moisture has some interesting applications. For example, in agriculture many research and private industry groups are using such routines from satellites and drones to schedule irrigation. However, the soil moisture may be more unconstrained in this case than can be suitable for reasonable management of irrigation amounts and timing. The authors could potentially comment on this application given the findings of the paper.

*This is certainly an interesting line of inquiry and one that our group is actively exploring via the use of UAVs in agricultural systems. However, as you suggest, we suspect that such approaches may not be particularly useful for irrigation scheduling. Rather, the techniques we are investigating*

*look to explore the spatial variability in crop systems, relating this back to spatially distributed areas of moisture stress or even over-application. While we anticipate that there should still be signs of coupling in these environments, the managed nature of the problem may make these links harder to disentangle. Given that we are dealing with quite a different problem in this particular semi-arid landscape example, we have not explored these interesting ideas in the present manuscript, but it is certainly an aspect worth future investigation.*

---

## Author Comment (AC3) · 19 Jul 2016

**Response to review comments (Round 2)**

*We very much appreciate the review comments from Dr. Bogena and thank him for the effort and attention to our manuscript. It has certainly been improved as a result. We have addressed all of the comments and present our responses below.*

*Our responses to the comments are indented and in italics.*

**Referee #1 (H. Bogena)**

What is important is the height of the EC sensors above canopy, which is only about 12 m in the case of Graf et al. (2014).

> *We agree, but in this and our particular case, the argument still stands that the footprint of an EC sensor depends on the height of the tower as well as certain prevailing meteorological considerations. For a 2 m tower height in a pasture (with low vegetation height), a fetch of around 200 m is a very reasonable estimate.*

An increasing number of existing large scale sensor networks make their data freely available to the science community (e.g. SCAN, ICOS). In addition, a number a measurement techniques are emerging that make use of existing networks that formally were installed for other reasons (e.g. Bogena et al., 2015) and thus will provide a much better coverage of soil moisture observation in the near future beyond the observatories.

> *We absolutely agree that CZO's and sensor networks are invaluable in providing data to understand the underlying processes behind hydrologic (and other related) variables and states – which we had stated in the text. However, while the number and coverage of such networks is increasing, it still remains impractical to have in-situ sensors at fine intervals of a few meters (or even hundred meters) to cover every area of interest. For example, intensively instrumenting an agricultural field would either create a hindrance to the farmer in his operations, or, on the other hand, result in the sensors being damaged or uprooted during tilling and other farming practices. We believe we have covered both aspects of this important area in our manuscript.*

I am still very much more in favor for adding this information. Why should the reader gather all this papers himself to get a basic overview of the models and their differences? Presenting this information makes the paper much more comprehensible and also better explains why the three models were used instead of only one.

> *As noted, we provided a basic overview of the models in the original manuscript. The premise (and common practice) of referencing to previously published technical articles is to avoid 1) inflating the length of the manuscript; 2) unnecessary duplication and 3) burdening the reader with material that can easily be obtained elsewhere (given the wide application of the chosen models).*

*However, we respect the reviewer's opinion and will provide some additional details regarding the models in a revised manuscript, while still maintaining references where the reader can obtain the more detailed model information.*

Actually, the CRNP validity was not tested in this paper in a strict sense. This could not be done with a single TDR profile anyway, since a network of point measurements within the CRNP footprint would be needed to do this (see e.g. Bogena et al., 2013).

*Our intention was only to use the TDR measurements to assess the CRNP retrievals i.e. they were not compared at a point-to-point level, but more from a general behavioral perspective, to ensure that there were no unreasonable patterns in the time series. We have changed the term validate to evaluate to reflect this more qualitative assessment.*

It is true that it was not implied that the observations were less accurate than the model results. The impression arises, because the focus was led on the comparison of soil moisture with model results.

*The objective of the study was to query the relationship between the CRNP-measured soil moisture and the modeled evaporation estimates. We will review the paper once again to ensure that any implication of relative accuracy is removed from the revised manuscript.*

First, I have to repeat again that the term "evaporation" is confusing. I guess you are referring to evapotranspiration because it is related to root zone (i.e. evaporation from bare soil and intercepted water is not related to root water uptake in the root zone). So again, please improve the terminology in the paper. What I was trying to point out is that the existence of similar distributions alone is not adequate for this deduction, because processes at the soil-vegetation-atmosphere interface tend to be very complex. For instance, the process of evaporation from canopy is completely independent from soil moisture, but it might be an important part of total evapotranspiration at this location. Therefore, the distributions of both quantities might be similar because the CRNP measurements are also influenced by water intercepted on the canopy (see e.g. Bogena et al., 2013). In addition, the CRNP typically does not cover the whole root zone, because for intermediate soil moisture ranges the penetration depth is restricted to 20-30 cm. In addition, the CNRS is much more sensible to soil moisture of the first centimeters. In this sense the TDR measurements might even better represent root zone soil moisture at this site. Thus, the Q-Q analysis should also be done with the TDR data to test the assumption soil moisture is driven by the evapotranspiration process.

*We are following the terminology established by Kalma et al., (2008), and employed widely elsewhere, wherein the term "evaporation" encompasses all processes resulting in transfer of water from the soil or vegetation to the atmosphere. As such, land surface evaporation consists of evaporation from the soil and canopy, and transpiration from the plant.*

*With regard to the Q-Q analysis, we agree that in some locations the CRNP can measure intercepted water, which could be a significant portion of the composite evaporation. It is worth noting that canopy interception is likely to represent an insignificant component of total evaporation in this semi-arid grassland environment.*

*However, as suggested, we performed a Q-Q analysis with the TDR measurements (see the plot below for the results). The blue + markers represent the analysis with the CRNP measurements, while the black dots represent the average of the three TDR probes. As can be seen, there is very little difference between the two plots for any of the model evaporation estimates.*

*We hope that this result satisfies the reviewer that our argument that the root zone soil moisture (RZSM) is still driving the evaporation process is valid. We have included text in the manuscript to reflect the logic put forth by the reviewer regarding TDR measurements perhaps being more representative of the RZSM, and that there was no significant difference in the Q-Q analyses with either CRNP or TDR measurements.*

[Figure]

I am sorry for my ignorance concerning the models used in this study. Clearly, a better description of the models will help the readers to be better following the reasoning presented in the paper. If soil moisture is not a model variable this should be explicitly mentioned in the paper. Otherwise the modelled soil moisture should be compared with the measured soil moisture to demonstrate the validity of the model.

*As mentioned in our earlier response, we will provide additional details about the models and have also revised the text to emphasize that soil moisture is not an input to any of the models evaluated in this study.*